# THE MATTHEW EFFECT OF AI PROGRAMMING ASSISTANTS: A HIDDEN BIAS IN SOFTWARE EVOLUTION

**Fei Gu**[1]*  **Zi Liang**[2]*  **Jiahao Ma**[3]  **Hongzong Li**[4]†
[1]City University of Hong Kong (Dongguan)
[2]The Hong Kong Polytechnic University
[3]The University of Hong Kong
[4]The Hong Kong University of Science and Technology

## ABSTRACT

AI-assisted programming is rapidly reshaping software development, with large language models (LLMs) enabling new paradigms such as vibe coding and agentic coding. While prior works have focused on prompt design and code generation quality, the broader impact of LLM-driven development on the iterative dynamics of software engineering remains underexplored. In this paper, we conduct large-scale experiments on thousands of algorithmic programming tasks and hundreds of framework selection tasks to systematically investigate how AI-assisted programming interacts with the software ecosystem. Our analysis quantifies a substantial performance asymmetry: **mainstream languages and frameworks achieve significantly higher success rates than niche ones**. This disparity suggests a feedback loop consistent with the **Matthew Effect**, where data-rich ecosystems gain superior AI support. While not the sole driver of adoption, current models introduce a non-negligible productivity friction for niche technologies, representing a hidden bias in software evolution.

## 1 INTRODUCTION

Large language models (LLMs) have quickly become ubiquitous in software engineering practice, with nearly all programmers (Daigle & Staff, 2024) utilizing AI coding tools. Tools such as GitHub Copilot, Cursor, and integrated LLM-based coding assistants now support developers in algorithmic problem solving (Yan et al., 2023), debugging, and even full-stack system construction. These advances introduce new coding paradigms: *vibe coding*, where developers iterate by prompting rather than typing every line, and *agentic coding*, where autonomous agents plan and execute end-to-end development tasks. Vibe coding democratizes software development (Gadde, 2025) by lowering barriers to creation, translating conceptual intent into executable implementation. Agent-based code generation highlights the transformative potential of multi-agent systems in addressing the limitations of standalone LLMs. Agentic Coding effectively handles real-world coding challenges (Wu et al., 2024) by leveraging external tools for retrieval, achieving significant improvements (Huang et al., 2023) in pass rates across diverse benchmarks (Zhang et al., 2024). Collectively, AI Coding could be the silver bullet for software engineering.

Prior to this empirical reality check, there was widespread optimism that LLMs would serve as a "Great Equalizer." Recent studies show that LLMs help narrow the skill gap for junior developers (Noy & Zhang, 2023; Metabob, 2024). However, the belief that they also function as a language equalizer, making specific syntax irrelevant as suggested by (Huang, 2024), has not yet been tested in empirical settings. We challenge this assumption and argue that instead of flattening the landscape, AI support introduces a critical new factor that may disproportionately disadvantage niche ecosystems.

Long-term ecosystem-level consequences of AI programming assistance remain underexplored. This research gap is critical because biases in training data and model behavior may systemat-

---

*Co-First Authors.
†Corresponding author: `lihongzong@ust.hk`.

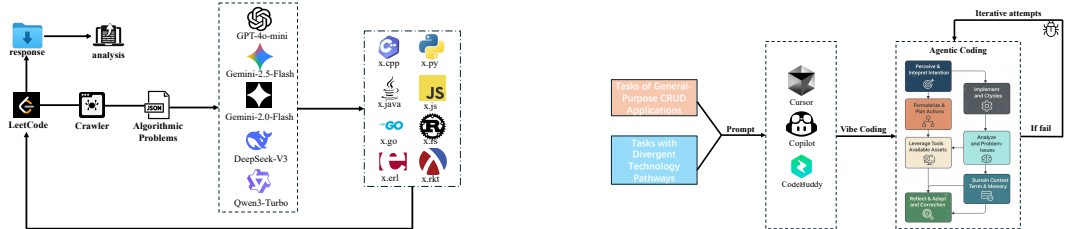

Figure 1: Two-tier experimental pipeline combining algorithmic tasks and framework tasks.

ically influence which languages, frameworks, and paradigms thrive or decline. Several observations underscore why this matters. First, LLM performance is uneven across languages: high-resource ecosystems such as Python achieve disproportionately strong results, while lower-resource languages receive much weaker support. For instance, the StarCoder dataset shows Python alone accounts for nearly 40% of its training corpus, while many other languages appear only marginally (Li et al., 2023). Similarly, CodeGen explicitly notes that model quality varies substantially depending on training data availability, with mainstream ecosystems benefiting disproportionately (Nijkamp et al., 2022). Second, the bias extends beyond languages to frameworks and usage patterns. AI coding assistants often over-rely on established libraries, such as NumPy, which appears in up to 48% of completions even when alternatives may be more suitable, and they also display a persistent preference for Python, which is selected 58% of the time for performance-critical tasks where other languages like Rust may be objectively better (Twist et al., 2025).Taken together, these patterns raise a central question: do AI tools genuinely empower innovation by lowering entry barriers, or do they inadvertently reinforce existing dominance hierarchies?

The hypothesis we explore is that AI programming assistants exhibit a **Matthew effect**: "the rich get richer." This effect is rooted in the operational mechanics of LLMs, which are trained on massive datasets of publicly available code. Such dynamics risk creating lock-in effects that suppress experimentation and reduce opportunities for paradigm-shifting innovations, which is consistent with prior observations on programming language adoption and diffusion (Meyerovich & Rabkin, 2013). Programming learners may increasingly favor languages where AI support is strongest, further consolidating existing trends (Prather et al., 2023). The empirical research on language adoption demonstrates that ecosystem factors (libraries, existing code, community size), rather than purely technical merit, strongly drive which languages gain traction, implying that model-mediated productivity gains could differentially amplify preexisting popularity patterns (Cao et al., 2022). Most existing studies of AI-assisted code generation focus on short-term, micro-level evaluations that measure model performance on narrow benchmarks or single-language datasets, which do not capture the multi-faceted complexity of real-world software engineering. If this impact is overlooked, the resulting cycle, where popular languages receive more LLM support due to their prevalence in training data, risks reducing programming ecosystem diversity. Thus, AI assistance could simultaneously lower barriers to entry while stifling long-term innovation.

**Contributions.** This paper makes three main contributions:

- We construct the first large-scale benchmark combining algorithmic programming tasks (Total $3011 \times 9 \times 5 = 135,495$) and complex full-stack development tasks to assess AI programming assistants across languages and frameworks.

- We design a controlled evaluation methodology that isolates the effect of language and framework popularity, revealing structural biases beyond aggregate accuracy metrics.

- We quantify a substantial asymmetry in AI support across languages and frameworks, demonstrating patterns consistent with Matthew-effect dynamics. While separating AI-specific amplification from pre-existing structural biases remains an open empirical question, our findings reveal a measurable "AI productivity tax" that correlates with ecosystem popularity.

Figure 1 presents a high-level overview of the two-tier experimental pipeline, illustrating both language-level algorithmic tasks and framework tasks

## 2 RELATED WORK

### 2.1 AI PROGRAMMING ASSISTANTS

Research on LLM-based programming has largely focused on improving code generation quality, prompt engineering, and usability. Systems like Codex and Copilot have demonstrated high productivity gains in everyday development. More recent models such as GPT-4, Gemini, and DeepSeek exhibit strong reasoning and multi-step planning abilities, further lowering the barrier to complex programming. Recent work has examined the capabilities of AI-assisted code generation tools across diverse benchmarks. Early evaluations using HumanEval (Chen et al., 2021) revealed that while Copilot often produced syntactically valid code, correctness rates were low and highly correlated with the prevalence of languages in the training data (Yetistiren et al., 2022). Although HumanEval became an early standard for evaluating LLM coding proficiency (Jiang et al., 2024), its limited number of problems restricts its applicability across all research contexts. To address this limitation, subsequent studies employed LeetCode problems. Copilot's best accuracy in Java (Nguyen & Nadi, 2022), subsequent analysis extended to multiple tools, reporting that Copilot excelled in Java, ChatGPT maintained strong cross-language consistency (Batista et al., 2024), and Gemini performed best in JavaScript. Larger-scale evaluations showed Copilot's accuracy decreased with problem difficulty and varied substantially across languages (Mo et al., 2025). The hyperscale multilingual benchmark-XCODEEVAL (Khan et al., 2024), demonstrated persistent challenges for program synthesis and translation, especially in less common languages, though its automatically collected data lacks manual verification and may introduce noise. Existing benchmarks offer valuable insights into AI code generation, but most focus on a few popular languages. By contrast, we use a standardized LeetCode-based dataset to examine how language popularity affects performance.

### 2.2 PROGRAMMING ECOSYSTEM EVOLUTION

The evolution of programming languages and frameworks has long been shaped by ecosystem factors such as community size, tooling, and industry adoption, often outweighing intrinsic technical merit (Meyerovich & Rabkin, 2013). In a longitudinal study of web framework popularity, Swacha & Kulpa (2023) demonstrated that adoption trajectories vary considerably across ecosystems. **AI as a New Adoption Factor:** The emergence of AI assistants introduces a complex new variable to this evolutionary dynamic. Recent literature presents a nuanced picture of AI-assisted development: while studies report substantial speedups on specific tasks (Peng et al., 2023), concerns regarding correctness, over-reliance, and workflow disruption persist (Weisz et al., 2024). This suggests that AI is not a simple linear accelerator but a nuanced influence on developer experience. However, recent surveys highlight a critical risk within this influence: LLM performance is uneven and heavily biased toward widely used languages (Zhang et al., 2023). Human-curated benchmarks such as CodeArena reveal persistent cross-language disparities, suggesting that LLMs may reinforce the dominance of mainstream ecosystems like Python and JavaScript (Yang et al., 2024). Beyond language adoption, ecosystem-level inequalities also emerge when LLMs act as autonomous coding agents: AgentBench (Liu et al., 2023) shows that proprietary models such as GPT-4 exhibit superior reasoning and multi-turn decision-making, while open-source models lag significantly. Together, these findings point to a risk that AI programming assistants could entrench existing hierarchies and amplify long-term ecosystem imbalances.

## 3 ENVIRONMENT AND BENCHMARK CONSTRUCTION

To evaluate AI programming assistants, we designed a benchmark focused on two core software engineering components: languages and frameworks. Using a controlled variable method, we first assess performance on algorithmic tasks across nine programming languages to see if popularity impacts success. We then evaluate the AI's ability to build real-world applications using various frameworks, testing both its proficiency on six mainstream full-stack combinations for common CRUD tasks and its architectural reasoning in specialized scenarios where niche technologies might be superior to popular ones. This dual approach allows us to measure not only core coding ability but also whether AI assistants exhibit a bias towards mainstream technologies, even when more suitable alternatives exist.

## 3.1 ALGORITHMIC TASKS

### 3.1.1 LANGUAGE SELECTION

We select nine languages guided (Table 1) by the June 2025 TIOBE Index (TIOBE Software BV, 2025): Python, C++, C, Java, JavaScript, Go, Rust, Erlang, and Racket. These cover a spectrum from top-ranked mainstream languages to niche or emerging languages, enabling examination of popularity effects. This design allows us to investigate whether language popularity correlates with the performance of AI-assisted code generation and submission success.

Table 1: Programming Languages Selected for Comparative Experiments and Their Popularity

| Language | TIOBE Rank (Jun 2025) | GitHub Repos (>100 stars) | Trend (5-yr) |
|---|---|---|---|
| Python | 1 | 185,000 | Strong Growth |
| C++ | 2 | 85,000 | Stable |
| C | 3 | 42,000 | Gradual Decline |
| Java | 4 | 125,000 | Gradual Decline |
| JavaScript | 6 | 172,000 | Stable |
| Go | 7 | 45,000 | Rapid Growth |
| Rust | 13 | 38,000 | Rapid Growth |
| Erlang | 46 | 1,200 | Declining |
| Racket | N/R | 450 | Niche/Stable |

### 3.1.2 TEST CASES FROM LEETCODE

To evaluate performance across these languages, we source algorithmic tasks from LeetCode (Leet-Code, 2025), an online platform chosen for its extensive collection of problems, robust online judging system, and broad multilingual support. LeetCode accommodates nearly 20 programming languages, including both widely used ones (C++, Java, Python) and less common ones (Rust, Scala, Elixir), making it an ideal environment for our multilingual study.

To assemble our dataset, we develop a script to systematically retrieve problem information via paginated POST requests to LeetCode's GraphQL endpoint. This process collects metadata such as problem titles, difficulty ratings, and tags, which are then processed into structured JSON files for analysis. We focus exclusively on publicly available, non-paid problems to ensure reproducibility.

Using this approach, we collect a total of **3,011 problems**, comprising **765 easy**, **1,526 medium**, and **720 hard** problems. This dataset forms the foundation for our large-scale benchmarking. To manage solution submission and validation, we employ a distributed submission system with 15 accounts, implementing exponential backoff and rate-limiting mechanisms to ensure scalable and reliable data collection.

We position these algorithmic tasks as the 'canary in the coal mine' for linguistic competency. While acknowledging that LeetCode does not capture the full software ecosystem, the high compile error rates observed in niche languages expose a fundamental deficiency. If an LLM cannot generate syntactically correct code for basic logic, this failure inherently precludes its effective application in broader and more complex engineering contexts.

## 3.2 FRAMEWORK SELECTION

Our evaluation employs a two-tiered benchmark designed to systematically assess LLM capabilities across different ecosystem contexts. The first tier, General-Purpose CRUD Applications, tests core code generation proficiency using six mainstream full-stack combinations selected by industry adoption metrics (GitHub stars, Stack Overflow activity, job postings), establishing a performance baseline in common development scenarios. The second tier, Tasks with Divergent Technology Pathways, examines model reasoning beyond popularity biases by presenting architectural trade-offs (e.g., performance vs. development speed), evaluating whether LLMs can identify and implement more suitable niche frameworks versus mainstream options. This structure enables a comprehensive assessment of both routine coding ability and adaptive architectural discernment.

Table 2: Full-stack Combinations Selected for Comparative Experiments and Their Popularity

| Stack | Components | GitHub Stars | Stack Overflow Tags | Job Description Frequency |
|---|---|---|---|---|
| Java Enterprise | Vue + Spring Boot + Hibernate | Vue: 200k+
Spring Boot: 78.4k
Hibernate: 460 | Vue: 100k+
Spring Boot: 100k+
Hibernate: 100k+ | High |
| Modern JS | React + Express.js + Prisma | React: 223k+
Express.js: 104k+
Prisma: 43.8k | React: 200k+
Express.js: 100k+
Prisma: Less Common | High |
| Python Full-stack | Django (REST) + Django ORM | Django: 85k+
DRF: 29.5k+ | Django: 100k+
DRF: 50k+ | High |
| Lightweight Go | Preact + Gin + GORM | Preact: 38k
Gin: 423
GORM: 38.8k | Preact: 10k+
Gin: 1k+
GORM: 10k+ | Medium |
| Modern Python | Svelte + FastAPI + SQLAlchemy | Svelte: 84.1k
FastAPI: 89.4k
SQLAlchemy: 10.9k | Svelte: 20k+
FastAPI: 20k+
SQLAlchemy: 50k+ | Medium-High |
| Rust Emerging | SolidJS + Actix Web + SeaORM | SolidJS: 34.2k
Actix Web: 23.3k
SeaORM: 8.7k | SolidJS: 5k+
Actix Web: 5k+
SeaORM: Less Common | Low |

To systematically evaluate LLM-assisted software development, we construct a benchmark consisting of five categories of tasks jointly designed by domain experts and industry practitioners. These tasks cover both generic development scenarios and cases with clear technology route divergences, allowing for a comprehensive evaluation of AI-assisted coding performance.

Our benchmark begins with a foundational set of **(1) Generic Tasks,** which includes 17 representative application scenarios frequently encountered in practice, such as movie ticket booking and library management systems. To ensure comparability across diverse ecosystems, each task is implemented across six mainstream full-stack frameworks, ranging from popular combinations like Vue with Spring Boot to emerging stacks like SolidJS with Actix, as detailed in Table 2. Building upon this baseline, the evaluation progresses to more specialized domains. For **(2) High-Concurrency Systems,** we assess tasks like real-time chat platforms, contrasting the mainstream Node.js/Socket.IO approach with the performance-oriented solutions offered by Go/-Gin and Rust/Actix. The framework then addresses **(3) Data-Intensive Applications,** using examples like log analytics to compare the dominant Python/Pandas ecosystem against enterprise-focused Scala/Spark and the niche scientific computing paradigm of Julia. Subsequently, to gauge performance in lower-level development, the fourth category focuses on **(4) Systems Infrastructure,** tasking the models with creating lightweight API gateways and distributed key-value stores using Go, Elixir/Phoenix, and Rust/Axum to cover popular, fault-tolerant, and emerging systems languages, respectively. Finally, the benchmark explores **(5) Alternative Programming Paradigms** by requiring declarative or functional solutions for services like chatbots, thereby comparing mainstream imperative languages with the distinct approaches of Haskell, Clojure, or F#.

The selected stacks span a wide spectrum: mainstream (Python, JavaScript, Java), emerging (Go, Rust, Kotlin), and niche (Elixir, Haskell, Clojure, Julia). This enables analysis not only of functional correctness but also of how LLMs handle underrepresented yet domain-relevant stacks. These frameworks are well-regarded in specific communities (e.g., concurrency, functional programming, scientific computing) but have limited adoption and significantly fewer resources in open-source datasets. This contrast allows us to measure not only whether the generated projects are executable but also whether LLMs disproportionately favor mainstream stacks, even when alternative stacks may be more suitable for the given task scenario.

## 3.3 Experimental Infrastructure

For both types of tasks, the same core methodology is applied. The variation in implementation arises merely from modifying the technology stack or paradigm specified in the prompt, while the functional requirements remain consistent. For the algorithmic tasks, the proprietary LLM APIs used are summarized in Appendix A.4. For the framework selection tasks, all work is performed

using three AI programming tools directly: Cursor Pro (using Claude-4-Sonnet), CodeBuddy (using Claude-4-Sonnet), and Visual Studio Code with GitHub Copilot (using GPT-5).

# 4 PROGRAMMING LANGUAGE ANALYSIS

## 4.1 AI CODING

For each of the 3,011 problems crawled from LeetCode, we apply a standardized procedure wherein the problem statement and constraints are formatted into a consistent prompt template (for each of the nine selected programming languages). In total, this process results in over 135,495 individual code generation requests (3,011 problems × 9 languages × 5 models), by calling the APIs of these five models: GPT-4o-mini (Hurst et al., 2024), DeepSeek-V3 (Liu et al., 2024), Gemini-2.0-Flash (Google, 2025), Gemini-2.5-Flash (Comanici et al., 2025), Qwen3-Turbo (Yang et al., 2025). This prompt is then submitted to each commercial closed-source LLM 's API to generate solutions.

Although we request that the AI generate pure code, its responses occasionally contained natural language text or other non-executable content. We specifically design the process to extract pure, executable code from mixed-text responses. This systematic approach ensures that the final output consists merely of functional code that can be directly submitted to LeetCode without any additional modifications, addressing the common challenge of irrelevant natural language explanations and cross-language code snippets in AI-generated content. By implementing a multi-stage cleaning process, the tool first identifies and extracts code blocks from potential Markdown formatting, then applies language-specific regular expression patterns to remove all forms of comments and non-code elements. The technical implementation employs targeted regular expression patterns tailored to each programming language's syntax characteristics, including '//.' and '/*.?*/' for C-style languages, '^#.\n?' for Python, '^%.\n?' for Erlang, and '^;.*\n?' for Racket. This language-aware approach effectively removes both single-line and multi-line comments while preserving code functionality. The refinement process additionally incorporates whitespace normalization and explanatory text filtration, resulting in clean, production-ready code that maintains the algorithmic integrity of the original AI-generated solution while eliminating all non-essential elements that would prevent immediate platform execution.

## 4.2 SOLUTION JUDGING

Each AI-generated solution is submitted without modifications to LeetCode's online judging system, with results systematically recorded for subsequent analysis. The platform categorizes submission outcomes into six distinct status types: Accepted, Compile Error, Wrong Answer, Runtime Error, Time Limit Exceeded and Memory Limit Exceeded. The primary evaluation metric is the **Pass@1** accuracy, defined as the fraction of solutions accepted on their first submission attempt.

To support this large-scale evaluation while respecting LeetCode's operational policies, we implement a distributed submission system utilizing multiple accounts with proper authentication mechanisms, including CSRF tokens and session cookies. The system incorporates an exponential backoff strategy with an initial 2-second delay and a maximum of 32 seconds for retries to gracefully manage HTTP 429 and other transient errors. Additionally, request throttling is enforced at a rate of 10 submissions per minute per account to prevent detection, avoid service disruption, and ensure ethical use of LeetCode's platform resources.

## 4.3 RESULTS

Our large-scale evaluation reveals a pronounced performance gap between popular and less popular programming languages, a disparity that is substantial and consistent across all five state-of-the-art models tested. As shown in Table 3, mainstream languages including Python, JavaScript, Java, C and C++ achieve Pass@1 rates exceeding 60% in top-performing models. In stark contrast, less popular languages such as Erlang and Racket struggle dramatically, with success rates often below 25% and sometimes approaching zero. For instance, the best-performing model (DeepSeek-V3) achieves 79.81% Pass@1 for Python but only 24.31% for Erlang and 20.82% for Racket. This pattern confirms that language popularity is a stronger predictor of AI coding success than model capability alone. This phenomenon, a pronounced Matthew effect in AI-assisted programming, becomes even

Table 3: Experimental Results across five LLMs and nine programming languages. Pass@1 denotes first-attempt success rate; error categories are reported as raw counts.

| Model | Lang | Pass@1 | Accepted | Wrong Ans. | Compile Err. | Runtime Err. | Other Err. | Easy | Medium | Hard |
|---|---|---|---|---|---|---|---|---|---|---|
| Gemini-2.5-Flash | Python | 67.92% | 2045 | 217 | 0 | 726 | 23 | 609 | 1104 | 331 |
| | C++ | 68.65% | 2067 | 164 | 744 | 13 | 23 | 617 | 1103 | 347 |
| | C | 58.59% | 1764 | 170 | 1007 | 52 | 18 | 572 | 925 | 267 |
| | Java | 68.65% | 2067 | 157 | 739 | 39 | 9 | 612 | 1106 | 349 |
| | JavaScript | 64.50% | 1942 | 275 | 0 | 781 | 13 | 592 | 1053 | 297 |
| | Go | 50.22% | 1512 | 105 | 1377 | 7 | 10 | 544 | 751 | 216 |
| | Rust | 51.81% | 1560 | 115 | 1311 | 17 | 82 | 507 | 837 | 216 |
| | Erlang | 1.26% | 38 | 4 | 2824 | 145 | 33 | 22 | 14 | 2 |
| | Racket | 17.10% | 515 | 94 | 2184 | 200 | 18 | 235 | 237 | 43 |
| Gemini-2.0-Flash | Python | 62.94% | 1895 | 268 | 0 | 787 | 61 | 594 | 1025 | 275 |
| | C++ | 64.26% | 1935 | 249 | 718 | 34 | 75 | 609 | 1048 | 278 |
| | C | 47.09% | 1418 | 304 | 1044 | 127 | 116 | 519 | 723 | 176 |
| | Java | 65.86% | 1983 | 273 | 652 | 42 | 61 | 603 | 1077 | 303 |
| | JavaScript | 64.40% | 1939 | 357 | 0 | 618 | 97 | 607 | 1057 | 275 |
| | Go | 55.90% | 1683 | 237 | 1088 | 17 | 66 | 527 | 907 | 249 |
| | Rust | 50.38% | 1517 | 267 | 1144 | 44 | 71 | 501 | 793 | 223 |
| | Erlang | 0% | 0 | 0 | 2918 | 93 | 0 | 0 | 0 | 0 |
| | Racket | 11.06% | 333 | 281 | 1995 | 350 | 52 | 180 | 139 | 14 |
| GPT-4o-mini | Python | 41.98% | 1265 | 444 | 0 | 1265 | 38 | 387 | 714 | 162 |
| | C++ | 41.68% | 1255 | 416 | 1234 | 72 | 34 | 428 | 683 | 144 |
| | C | 38.43% | 1157 | 460 | 1169 | 191 | 34 | 474 | 518 | 165 |
| | Java | 45.50% | 1370 | 452 | 1055 | 99 | 35 | 476 | 746 | 148 |
| | JavaScript | 45.57% | 1372 | 564 | 0 | 1031 | 44 | 495 | 738 | 139 |
| | Go | 39.22% | 1181 | 405 | 1375 | 18 | 32 | 451 | 622 | 108 |
| | Rust | 24.05% | 724 | 322 | 1915 | 29 | 21 | 308 | 359 | 57 |
| | Erlang | 1.16% | 35 | 77 | 2701 | 195 | 6 | 27 | 6 | 2 |
| | Racket | 1.99% | 60 | 147 | 2661 | 131 | 12 | 37 | 22 | 1 |
| Qwen3-Turbo | Python | 37.00% | 1114 | 401 | 0 | 1117 | 54 | 405 | 610 | 99 |
| | C++ | 30.22% | 910 | 411 | 1608 | 68 | 14 | 367 | 462 | 81 |
| | C | 21.65% | 652 | 439 | 1758 | 141 | 21 | 310 | 306 | 36 |
| | Java | 32.55% | 980 | 118 | 1886 | 19 | 8 | 337 | 491 | 151 |
| | JavaScript | 38.63% | 1163 | 618 | 0 | 1196 | 34 | 450 | 616 | 97 |
| | Go | 33.15% | 998 | 403 | 1566 | 26 | 18 | 388 | 533 | 77 |
| | Rust | 2.19% | 66 | 22 | 2915 | 7 | 1 | 29 | 33 | 4 |
| | Erlang | 0% | 0 | 0 | 2873 | 138 | 0 | 0 | 0 | 0 |
| | Racket | 3.25% | 98 | 201 | 2505 | 176 | 31 | 59 | 38 | 1 |
| DeepSeek-v3 | Python | 79.81% | 2403 | 0 | 162 | 418 | 28 | 683 | 1294 | 426 |
| | C++ | 78.81% | 2373 | 450 | 133 | 28 | 27 | 674 | 1279 | 420 |
| | C | 67.78% | 2041 | 268 | 497 | 122 | 83 | 668 | 1071 | 302 |
| | Java | 79.38% | 2390 | 412 | 152 | 29 | 28 | 681 | 1288 | 421 |
| | JavaScript | 75.69% | 2279 | 0 | 230 | 469 | 33 | 671 | 1227 | 381 |
| | Go | 76.82% | 2313 | 497 | 150 | 16 | 35 | 673 | 1249 | 391 |
| | Rust | 71.24% | 2145 | 625 | 199 | 22 | 20 | 644 | 1161 | 340 |
| | Erlang | 24.31% | 732 | 1445 | 373 | 396 | 65 | 378 | 321 | 33 |
| | Racket | 20.82% | 627 | 1805 | 287 | 197 | 95 | 326 | 268 | 33 |

more dramatic when stratified by problem difficulty. As illustrated in Figure 2, the performance gap widens substantially as complexity increases. For Easy problems, the difference between popular and niche languages ranges from 45 to 82 percentage points. This gap expands significantly to 58 to 95 points on Hard problems, indicating that the advantage of data-rich languages scales non-linearly with reasoning complexity. On these Hard tasks, top models achieve 50 to 63% success with popular languages but only 0 to 6% with less popular ones, demonstrating that superior model capability cannot compensate for the disadvantage of language unpopularity.

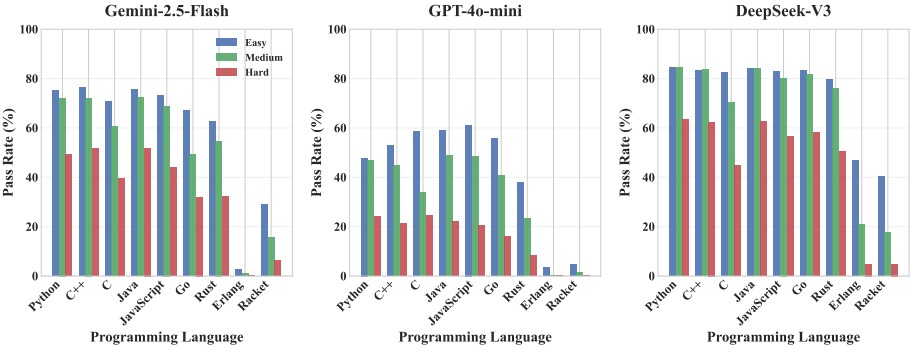

Figure 2: Pass rates across difficulty levels for top LLMs on nine programming languages.

Beyond success rates, the distribution of failure types reveals the mechanistic basis of this effect. For popular languages, most unsuccessful submissions are Wrong Answer or Runtime Errors, suggesting models generate semantically plausible but incorrect solutions. By contrast, failures in low-resource languages are dominated by Compile Errors, indicating models struggle to produce even syntactically valid code. This points to a deeper structural limitation: insufficient training exposure hinders the ability of models to internalize basic coding idioms. To ensure these differences are not due to random variation, we conducted paired t-tests comparing Pass@1 rates. As summarized in Table 4, the differences were statistically significant across all models ($p < 0.001$), confirming that the observed performance gaps reflect systematic biases.

Table 4: Statistical significance of Pass@1 differences between popular languages (Python, C++, C, Java, JavaScript) and less popular languages (Go, Rust, Erlang, Racket). All tests used paired t-tests across the 3,011 problems.

| Model | Mean Difference (%) | p-value |
|---|---|---|
| DeepSeek-V3 | +44.8 | $< 0.001$ |
| Gemini-2.5-Flash | +42.3 | $< 0.001$ |
| Gemini-2.0-Flash | +40.5 | $< 0.001$ |
| GPT-4o-mini | +33.1 | $< 0.001$ |
| Qwen3-Turbo | +28.9 | $< 0.001$ |

The observed Matthew effect has profound implications for programming language ecosystems. As AI-assisted programming becomes pervasive, the massive performance advantage for popular languages may accelerate their dominance while marginalizing niche languages, regardless of their technical merits. This could ultimately reduce linguistic diversity in software development. Our study uniquely leverages "data contamination" as a direct signal of language popularity, defining the overlap between test tasks and training data as a measure of a language's representation in the training corpus. This premise dictated our deliberate selection of newly released LeetCode tasks from 2025. This choice minimizes "rote recall" from widely-circulated problems and aligns the "contamination gap" with contemporary popularity trends, allowing us to establish a clearer causal chain: language popularity dictates training data coverage, which in turn drives AI performance.

## 5 FRAMEWORK ANALYSIS

After the evaluation of programming languages, we extend our study to software frameworks, which represent higher-level abstractions shaping developer workflows. Unlike languages, frameworks bundle architectural choices and toolchains, making them a crucial layer where LLM biases may influence ecosystem trajectories. Our analysis therefore examines whether a similar Matthew effect appears at the framework level, and to what extent mainstream stacks enjoy disproportionate advantages over niche alternatives.

### 5.1 VIBE CODING

For each development task, the implementation process across varying technology stacks followed a rigorously controlled VibeCoding protocol using the Cursor(Claude-4-Sonnet), CodeBuddy (Claude-4-Sonnet), and Copilot (GPT-5) in both Agent Mode (for high-level planning and multi-file generation) and Auto Mode (for inline code completion and contextual suggestions). The process commenced with an initial prompt that specified the functional requirements of the task along with the designated technology stack, no other contextual or syntactic guidance was provided. Throughout the implementation, the experimenter abstained from any manual coding, architectural input, or corrective intervention. The interaction was strictly limited to forwarding raw, unedited error messages, whether from dependency installation, compilation, runtime execution, or functional shortcomings, back into the chat interface as successive prompts. Each error message initiated a new, automated debugging attempt by the agent, continuing in an iterative loop without additional human elaboration. The procedure terminated only when all core functional requirements were satisfactorily met and the application operated as intended, or when a predetermined cap on iterative attempts was reached. This approach ensured that the observed outcomes were attributable solely to the AI's autonomous capacity to reason about and implement solutions within each technological context.

The empirical results provide strong evidence of a Matthew Effect in programming framework adoption under AI-assisted coding. Specifically, the success rate and efficiency of code generation were strongly skewed toward a few dominant frameworks. For instance, Vue+Spring, React+Express, and Django consistently solved the majority of the 17 benchmark tasks, often within 1–3 attempts. In contrast, less prevalent frameworks such as Svelte+FastAPI and SolidJS+Actix exhibited far higher failure rates; many tasks required more than five attempts or could not be completed at all.

The heatmap analysis (Figure 3) highlights this disparity. Successful completions clustered around the established frameworks, while newer or niche stacks displayed darker regions (representing repeated failures). Importantly, this pattern emerged across all categories of tasks, from lightweight personal applications (e.g., a birthday reminder tool) to more complex management systems (e.g., library management or inventory control). This consistency suggests that the observed bias is not task-specific but structural.

## 5.2 RESULTS

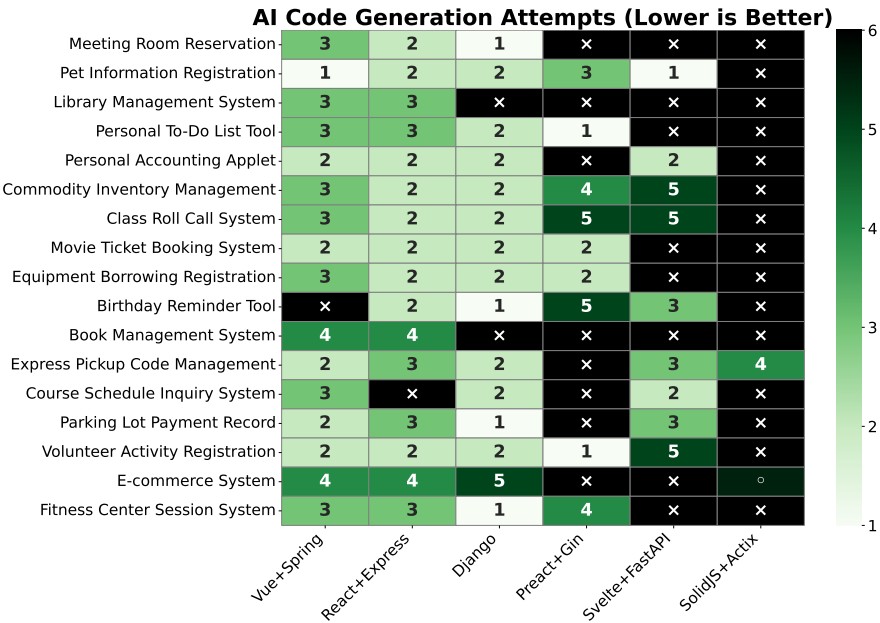

Figure 3: performance of 17 program tasks under 6 technical frameworks

Moreover, even in tasks where less popular stacks are technically well-suited (e.g., high-concurrency systems where Go or Rust frameworks should excel), the models still disproportionately favored Python- and JavaScript-based solutions. This demonstrates that the disparity arises not purely from technical merit, but from the underlying training data distribution. Frameworks with richer online presence and broader community adoption provide significantly more exposure during model pre-training, resulting in higher generation quality and stability.

Taken together, these findings illustrate a self-reinforcing feedback loop with significant implications for software ecosystem diversity: popular frameworks are easier for LLMs to generate successfully, developers relying on AI assistants are nudged toward these frameworks, and increased adoption further amplifies their online presence, ensuring even more model exposure in future iterations. Such dynamics exemplify the Matthew Effect in software ecosystems, where established technologies "get richer" in terms of visibility, usability, and adoption. While convenient for practitioners using mainstream stacks, this trend risks stifling ecosystem diversity by systematically disadvantaging technically promising but less popular frameworks. The findings further reveal that framework maturity and ecosystem support significantly impact AI code generation, with emerging frameworks lagging behind, suggesting that LLM-based assistance could exacerbate adoption gaps between established and new technologies.

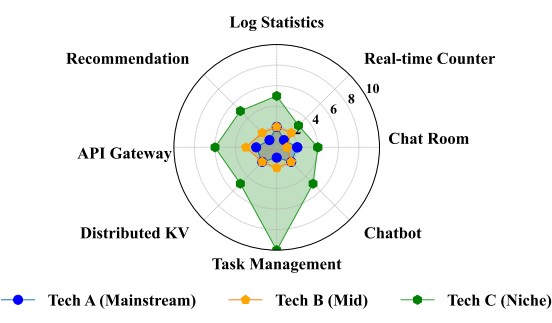

Figure 4: Results of Divergent Technology Pathway Benchmarks

As shown in Figure 4, the experiments of divergent-technology-pathway reinforce this conclusion. In scenarios explicitly designed to pit mainstream, middle-ground, and niche stacks against each other, for example, API gateways, distributed key-value stores, or chatbot systems, the number of required human interventions diverged sharply. Mainstream stacks (A) typically converged in 1–2 correction rounds, middle-ground stacks (B) required slightly more effort (2–3 interventions), while niche or emerging stacks (C) often demanded 5–10 rounds of guidance before producing a runnable system. These results confirm that even when controlling for task type, ecosystem popularity heavily conditions the reliability of AI-generated code.

# 6 CONCLUSION

This study provides the first large-scale empirical evidence of the Matthew effect in AI programming assistants, demonstrating how LLMs systematically amplify existing popularity hierarchies among programming languages and frameworks. Our findings reveal that mainstream technologies consistently achieve higher success rates in code generation, while niche and emerging alternatives face disproportionate failure rates that could potentially stifle innovation. We emphasize that technical decision-making is multi-dimensional; AI compatibility is not a universal veto that overrides established factors like runtime performance. However, our results quantify a substantial 'AI Productivity Tax' for niche languages. This creates a hidden friction consistent with Matthew-effect dynamics, which may disproportionately influence new projects and long-term ecosystem diversity. Moving forward, we plan to expand our benchmarks into broader domains, investigate collaborative multi-agent development scenarios, and develop methods to counteract ecosystem homogenization through diversity-aware training and inference strategies.

## REPRODUCIBILITY STATEMENT

We ensure reproducibility by releasing the complete benchmark dataset, prompt templates, and evaluation code in our public repository at `https://github.com/FrankGGu/The-Matthew-Effect-of-AI-Programming-Assistants`. Specifically, the implementation for algorithmic tasks (LeetCode retrieval/submission and result logging) is provided under `leetcode/`, while the framework-task prompts and related materials are under `task prompt/`. The benchmark composition is detailed in Appendix A, and the prompt design as well as the multi-stage code-extraction pipeline are described in Appendix B. Our experimental infrastructure is documented in Section 3.3, and we summarize the proprietary LLM APIs in Table 6. For artifacts that exceed repository size limits (e.g., raw logs or large generated projects), we distribute them via GitHub Releases associated with the same repository.

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

APPENDIX

LLM USAGE STATEMENT

This study **used large language models (LLMs)** as a core tool within the experimental process to **generate and evaluate code samples**. Our experiments involved calling the APIs of multiple LLM models (including GPT-4o-mini, DeepSeek-V3, Gemini-2.0-Flash, Gemini-2.5-Flash, and Qwen3-Turbo), making a total of over 135,495 code generation requests. These models were used to generate solutions for thousands of algorithmic programming tasks and hundreds of framework selection tasks, systematically investigating how AI-assisted programming impacts the software ecosystem. We followed a strict VibeCoding protocol, where the LLMs acted as autonomous agents to produce runnable code through iterative feedback on error messages.

To be clear, while LLMs were an integral part of the experimental process in this study, **they were not used to generate the research ideas, experimental design, or data analysis** for this paper. These aspects were performed independently by the authors.

ETHICS STATEMENT

This study relies exclusively on data from **LeetCode.cn**, a platform independently operated by **Lingkou Network (Shanghai) Co., Ltd.** and governed by its own Terms of Service (https://leetcode.cn/terms-c). The intellectual-property attorney confirmed that retrieving publicly available problem metadata, without bypassing technical protection, while respecting the Robots protocol, and for non-commercial academic research, is compliant with both Chinese law and the LeetCode.cn Terms of Service. We additionally confirmed our access pattern and usage constraints with LeetCode.cn staff.

All accounts used for submitting model-generated solutions were manually created, used solely for distributing evaluation load, and operated strictly within normal rate limits. We did not attempt to obtain platform benefits, circumvent protections, or access any non-public content. We do not process, collect, or store personal data, user-generated content, or paid materials from LeetCode.cn. We do not redistribute proprietary problem statements or other restricted content. Researchers can reproduce our evaluation pipeline using authenticated LeetCode web endpoints (problem retrieval via GraphQL and solution submission via standard session-based requests), subject to platform rate limits and Terms of Service.

## A    BENCHMARK SPECIFICATIONS

### A.1    LEETCODE BENCHMARK COMPOSITION

The LeetCode benchmark used in this study comprises 3,011 programming problems collected from the platform. The dataset contains 765 Easy (25.4%), 1,526 Medium (50.7%), and 720 Hard (23.9%) problems, representing a standard ordinal structure (Ma et al., 2026) that allows for fine-grained performance analysis.

Figure 5 presents the distribution of the top 15 algorithmic topics by difficulty level. The most prevalent tags include Array (1,777 problems), String (737 problems), and Hash Table (638 problems). Notably, Dynamic Programming problems are predominantly Medium (270) and Hard (287) difficulty, reflecting the challenging nature of this topic. Conversely, Two Pointers and Math problems show stronger representation in the Easy and Medium categories.

### A.2    DATA AVAILABILITY AND REPRODUCIBILITY

We release the benchmark construction scripts, prompt templates, evaluation pipeline, and analysis code at: https://github.com/FrankGGu/The-Matthew-Effect-of-AI-Programming-Assistants. The experiment is fully reproducible using the provided codebase, though we note two practical constraints: (1) the evaluation requires 7–10 weeks due to rate limiting and scale (135,495 code generations), and (2) reproduction incurs significant API costs ($1800-$2,000 USD).

**Distribution of Top Algorithmic Topics Across Difficulty Levels**

| Topic | Easy | Medium | Hard | Total |
|---|---|---|---|---|
| Prefix Sum | 20 (10.5%) | 116 (61.1%) | 54 (28.4%) | Total: 190 |
| Two Pointers | 69 (34.0%) | 109 (53.7%) | 25 (12.3%) | Total: 203 |
| Tree | 49 (21.8%) | 125 (55.6%) | 51 (22.7%) | Total: 225 |
| Breadth-First Search | 25 (10.7%) | 143 (61.1%) | 66 (28.2%) | Total: 234 |
| Bit Manipulation | 51 (21.8%) | 103 (44.0%) | 80 (34.2%) | Total: 234 |
| Matrix | 39 (16.2%) | 134 (55.8%) | 67 (27.9%) | Total: 240 |
| Binary Search | 36 (12.8%) | 144 (51.1%) | 102 (36.2%) | Total: 282 |
| Depth-First Search | 46 (15.8%) | 172 (58.9%) | 74 (25.3%) | Total: 292 |
| Greedy | 48 (12.8%) | 243 (64.8%) | 84 (22.4%) | Total: 375 |
| Sorting | 90 (21.5%) | 240 (57.3%) | 89 (21.2%) | Total: 419 |
| Math | 156 (28.2%) | 251 (45.3%) | 147 (26.5%) | Total: 554 |
| Dynamic Programming | | 270 (47.0%) | 287 (49.9%) | Total: 575 |
| Hash Table | 167 (26.2%) | 366 (57.4%) | 105 (16.5%) | Total: 638 |
| String | 219 (29.7%) | 352 (47.8%) | 166 (22.5%) | Total: 737 |
| Array | 395 (22.2%) | 925 (52.1%) | 457 (25.7%) | Total: 1777 |

Percentage Distribution by Difficulty (%)

Easy — Medium — Hard

Figure 5: Distribution of top algorithmic topics across difficulty levels, showing the number and percentage of problems for each difficulty category

## A.3    DETAILED DESCRIPTION OF THE 17 GENERAL-PURPOSE CRUD TASKS

The table 5 provides a detailed description of the 17 general-purpose application scenarios used to evaluate framework selection in Section 5. These tasks were designed to cover a wide range of common software development requirements while maintaining a comparable level of complexity.

## A.4 PROPRIETARY LLM APIS

Table 6 summarizes the proprietary large language models used in our algorithmic task experiments. We include details on version identifiers, release dates, and knowledge cut-off points to ensure reproducibility and clarify the temporal alignment between training data and evaluation tasks.

## B    PROMPT ENGINEERING & CODE EXTRACTION METHODOLOGY

### B.1    STANDARDIZED PROMPT DESIGN AND IMPLEMENTATION

#### B.1.1    PRIMARY PROMPT TEMPLATE STRUCTURE

```python
def _generate_prompt(self, problem_data: Dict) -> str:
    title = problem_data.get('title', '')
    description = problem_data.get('description', '')

    prompt = f"""
    Provide a {self.language_name} solution for LeetCode problem
        '{title}'.
```

| Task Name | Core Functional Requirements |
|---|---|
| E-commerce System | Product CRUD, recommendation logic, order creation/status workflow. |
| Library Management System | Search books, track loans, calculate overdue fines, category statistics. |
| Personal To-Do List Tool | Add, mark complete/delete, and filter todo items by date. |
| Meeting Room Reservation System | Display rooms, reserve/check availability in time slots, view bookings. |
| Personal Accounting Applet | Record income/expense (amount, category, note), view monthly summary. |
| Commodity Inventory Management | Add products (name, specs), update stock levels, log inbound/outbound. |
| Class Roll Call System | Manage student list, record daily attendance (present/absent), count absences. |
| Pet Information Registration | Register pets (name, breed, owner), log vaccination records. |
| Movie Ticket Booking System | Display movie schedules, select seats, generate and manage orders. |
| Equipment Borrowing Registration | Manage equipment (name, model), record borrow/return dates and user. |
| Birthday Reminder Tool | Add contacts and birthdays, list upcoming birthdays for the month. |
| Express Pickup Code Management | Enter parcel info (tracking #, code, recipient), mark as picked up. |
| Course Schedule Inquiry System | Maintain course info (name, teacher, room), display weekly/daily schedule. |
| Parking Lot Payment Record | Record vehicle entry/exit timestamps, calculate fee based on duration. |
| Volunteer Activity Registration | Publish activities (name, time, location), register users, count participants. |
| Book Management System | Register books (title, donor, ISBN), record borrowing history. |
| Fitness Session Consumption | Manage user membership cards, deduct sessions on check-in, check balance. |

Table 5: Detailed description of the 17 General-Purpose CRUD application tasks used for framework evaluation.

Table 6: Summary of proprietary LLM APIs used in Algorithmic tasks.

| Model | Version / Endpoint | Release | Knowledge Cut-off |
|---|---|---|---|
| GPT-4o-mini | GPT-4o-mini-2024-07-18 | Jul 2024 | Oct 2023 |
| DeepSeek-V3 | DeepSeek-V3-0324 | Mar 2025 | Mar 2025 |
| Gemini-2.0-Flash | Gemini-2.0-flash-001 | Feb 2025 | Jun 2024 |
| Gemini-2.5-Flash | Gemini-2.5-flash | Jun 2025 | Jan 2025 |
| Qwen3-Turbo | Qwen-turbo-2025-04-28 | Apr 2025 | Apr 2025 |

```
IMPORTANT: Do not think through the problem step by step. Just
    provide the code directly.
Code requirements:
1. Must compile and run on LeetCode
2. No comments or explanations
3. Only the solution code

Problem Description:
{description}

Code:
"""
return prompt
```

### B.1.2 SYSTEM-LEVEL INSTRUCTION CONFIGURATION

```
"systemInstruction": {
    "parts": [
        {"text": "You are a code generation assistant. Provide only code
            without any explanations or thinking process. Do not think
            through problems step by step."}
    ]
}
```

## B.2 MULTI-STAGE CODE EXTRACTION PIPELINE

### B.2.1 CODE BLOCK BOUNDARY IDENTIFICATION

```
# Code block start pattern detection
start_patterns = [
    r"```\w*\n",          # Standard code block markers ('''rust\n)
    r"```\n",             # Generic code blocks ('''\n)
    r"'''.*?\n",          # Python multiline string markers
    r'""".*?\n',          # Python multiline string markers
]

# Code block end pattern detection
end_patterns = [
    r"\n```\s*$",         # Standard end markers
    r"\n```\s*\n",        # End markers with newlines
    r"\n'''\s*$",         # Python end markers
    r'\n"""\s*$',         # Python end markers
]
```

### B.2.2 LANGUAGE-SPECIFIC SYNTAX CLEANING

```
# Python: Remove # comments
if self.language in ["python3"]:
    solution = re.sub(r'^#.*\n?', '', solution, flags=re.MULTILINE)
# C-family languages: Remove // and /* */ comments
elif self.language in ["cpp", "java", "javascript", "go", "rust"]:
    solution = re.sub(r'^//.*\n?', '', solution, flags=re.MULTILINE)
    solution = re.sub(r'/\*.*?\*/', '', solution, flags=re.DOTALL)
# Erlang: Remove % comments
elif self.language == "erlang":
    solution = re.sub(r'^%.*\n?', '', solution, flags=re.MULTILINE)
# Racket: Remove ; comments
elif self.language == "racket":
    solution = re.sub(r'^;.*\n?', '', solution, flags=re.MULTILINE)
```

### B.2.3 DEBUGGING ARTIFACT REMOVAL

```
# Remove debugging statements and test code
solution = re.sub(r'console\.(log|warn|error|info)\(.*?\);?\s*', '',
    solution)
solution = re.sub(r'function\s+test\w*\s*\(.*?\)\s*{[\s\S]*?\n}', '',
    solution)
solution = re.sub(r'document\..*?;', '', solution)  # Remove DOM
    operations
```

### B.2.4 CODE QUALITY OPTIMIZATION

```
# Remove excessive blank lines and explanatory text
solution = re.sub(r'\n\s*\n', '\n\n', solution)  # Compress blank lines
solution = re.split(r'\n(?:This code|Code
    explanation|Explanation|note:|Note:)',
                    solution, flags=re.IGNORECASE)[0]  # Remove trailing
                        explanations
```

### B.3 ROBUST ERROR HANDLING FRAMEWORK

#### B.3.1 EXPONENTIAL BACKOFF RETRY STRATEGY

```
max_retries = 5
base_delay = 2
max_delay = 30

# Exponential backoff + random jitter
delay = min(base_delay * (2 ** retry_count), max_delay)
jitter = random.uniform(0, 1)
sleep_time = delay + jitter
```

#### B.3.2 ERROR CLASSIFICATION AND HANDLING

```
if response.status_code == 200:
    # Success case processing
    return parts[0]['text'], None
elif response.status_code == 429:
    # Rate limiting - retry with backoff
    error_msg = f"Rate limited: {response.status_code}"
elif response.status_code >= 400 and response.status_code < 500:
    # Client error - do not retry
    error_msg = f"Client error: {response.status_code}"
else:
    # Server error - retry with backoff
    error_msg = f"Server error: {response.status_code}"
```

This comprehensive methodology framework ensured consistent, high-quality code generation across all nine programming languages while maintaining robustness through systematic error handling and validation procedures. The multi-stage extraction pipeline guaranteed that generated solutions met LeetCode's strict requirements for executable, comment-free code submissions.

### B.4 EXAMPLE INITIAL PROMPTS FOR FULL-STACK TASKS

Example initial prompt for the Movie Booking System task using the Modern JS stack:

```
Build a complete movie ticket booking web application using React for the
frontend, Express.js for the backend, and Prisma with SQLite for the
    database.
The application should allow users to browse movies, view showtimes,
    select
seats, and complete a booking. Provide the complete code.
```

### B.5 FRAMEWORK TASK PROMPT STRUCTURE WITH EXAMPLE

This section details the prompt structure used for the 17 general-purpose full-stack development tasks evaluated in this study. All tasks followed the same prompt pattern: a detailed specification of functional requirements and database schema, followed by instructions for the specific technology stack to be used.

To illustrate this structure, we provide the complete prompt for the *Meeting Room Booking System* as a representative example. The prompts for the other 16 tasks followed an identical format, with their respective requirement specifications substituted accordingly.

#### B.5.1 EXAMPLE: MEETING ROOM BOOKING SYSTEM

```
Database schema:
- users: id, username, email, full_name, department, created_at
- meeting_rooms: id, name, location, capacity, amenities, is_active,
    created_at
```

```
- bookings: id, room_id, user_id, title, description, start_time,
    end_time, status, created_at, updated_at

Requirements:
1. User Management:
    - JWT-based authentication
    - User profile management
    - Department-based organization

2. Meeting Room Management:
    - CRUD operations for meeting rooms
    - Availability checking
    - Filtering by various criteria

3. Booking Management:
    - Create, view, update, delete bookings
    - Time conflict detection
    - Status management
```

### B.5.2 TECHNOLOGY STACK VARIATIONS

For each of the six technology stacks evaluated, only the TECHNOLOGY STACK portion of the prompt was modified while keeping the requirements identical. The specific stack instructions were:

**Vue + Spring Boot + Hibernate (Java Enterprise):**

```
Create a complete meeting room booking system using Vue 3 for frontend
    and Spring Boot with Hibernate for backend. Use PostgreSQL database
    with the following schema:
Technical Specifications:
- Frontend: Vue 3 with Composition API, Vue Router for navigation, Pinia
    for state management
- Backend: Spring Boot with Spring Security for JWT authentication,
    Hibernate for ORM
- Database: PostgreSQL with connection string:
    postgresql://postgres:meetingpass@localhost:5432/meeting_booking
- API: RESTful design with proper HTTP status codes
Generate complete, runnable code including:
- Spring Boot application with controllers, services, and repositories
- Vue 3 components for all features
- Proper error handling and validation
- Database configuration and entity classes
- Installation and setup instructions
```

**React + Express.js + Prisma (Modern JS):**

```
Develop a meeting room booking system using React 18 for frontend and
    Express.js with Prisma for backend. Use PostgreSQL database with the
    following schema:
Technical Specifications:
- Frontend: React 18 with functional components and hooks, React Router
    for navigation
- Backend: Express.js with JWT authentication, Prisma as ORM
- Database: PostgreSQL with connection string:
    postgresql://postgres:meetingpass@localhost:5432/meeting_booking
- API: RESTful endpoints with proper error handling
Generate complete, runnable code including:
- Express.js server with routes, middleware, and controllers
- React components with modern hooks
- Prisma schema and migrations
- Authentication system
- Setup and deployment instructions
```

**Django REST Framework + Django ORM (Python Full-stack):**

```
Create a meeting room booking system using Django REST Framework for
    backend and a modern JavaScript framework for frontend. Use
    PostgreSQL database with the following schema:
Technical Specifications:
- Backend: Django with Django REST Framework, Django ORM
- Frontend: Use a modern JavaScript framework (specify which one)
- Database: PostgreSQL with connection string:
    postgresql://postgres:meetingpass@localhost:5432/meeting_booking
- API: RESTful design with token authentication
Generate complete, runnable code including:
- Django models, serializers, views, and URLs
- Frontend components and API integration
- Authentication system
- Database migrations
- Setup and running instructions
```

### Preact + Gin + GORM (Lightweight Go):

```
Build a lightweight meeting room booking system using Preact for
    frontend and Gin with GORM for backend. Use PostgreSQL database with
    the following schema:
Technical Specifications:
- Frontend: Preact with lightweight state management
- Backend: Gin framework with JWT authentication, GORM as ORM
- Database: PostgreSQL with connection string:
    postgresql://postgres:meetingpass@localhost:5432/meeting_booking
- API: RESTful design with minimal overhead
Generate complete, runnable code including:
- Gin server with routes, middleware, and handlers
- Preact components with minimal dependencies
- GORM models and database operations
- Authentication system
- Build and run instructions
```

### Svelte + FastAPI + SQLAlchemy (Modern Python):

```
Develop a modern meeting room booking system using Svelte for frontend
    and FastAPI with SQLAlchemy for backend. Use PostgreSQL database
    with the following schema:
Technical Specifications:
- Frontend: Svelte with SvelteKit for routing
- Backend: FastAPI with SQLAlchemy as ORM, Pydantic for validation
- Database: PostgreSQL with connection string:
    postgresql://postgres:meetingpass@localhost:5432/meeting_booking
- API: RESTful design with OpenAPI documentation
Generate complete, runnable code including:
- FastAPI application with routes, models, and schemas
- Svelte components with reactive programming
- SQLAlchemy models and database operations
- Authentication system with JWT
- Setup and running instructions
```

### SolidJS + Actix Web + SeaORM (Rust Emerging):

```
Create a meeting room booking system using the emerging Rust stack:
    SolidJS for frontend and Actix Web with SeaORM for backend. Use
    PostgreSQL database with the following schema:
Technical Specifications:
- Frontend: SolidJS with fine-grained reactivity
- Backend: Actix Web with JWT authentication, SeaORM as ORM
- Database: PostgreSQL with connection string:
    postgresql://postgres:meetingpass@localhost:5432/meeting_booking
- API: RESTful design with focus on performance
Generate complete, runnable code including:
- Actix Web server with routes, handlers, and middleware
```

```
- SolidJS components with reactive patterns
- SeaORM entities and database operations
- Authentication system with JWT
- Build and run instructions for both frontend and backend
```

### B.6    PROMPTS FOR EXPERIMENTS WITH DIVERGENT TECHNOLOGY ROUTES

The following prompts were used to instruct the AI coding tool. Eight experiments were designed to emphasize typical technical route divergences, covering scenarios such as API gateways, chat servers, data pipelines, task queues, GraphQL services, event streaming, edge inference, and blockchain explorers. Each task specifies three alternative stacks (A, B, C) representing distinct trade-offs in performance, ecosystem maturity, and adoption trends.

Listing 1: Experiment 1: High-Concurrency API Gateway

```
Task: Build a high-concurrency API gateway that forwards requests to
    backend services and supports basic rate limiting.

Technology stack (must use exactly this):
[Option A] Rust + Axum + Tokio
[Option B] Go + Gin
[Option C] Python + FastAPI + Uvicorn

Requirements:
- Accept HTTP requests on /api.
- Forward requests to a mock backend service.
- Implement simple rate limiting per client IP.
```

Listing 2: Experiment 2: Real-Time Chat Server

```
Task: Build a simple chat server where clients can connect and send
    messages to each other.

Technology stack (must use exactly this):
[Option A] Elixir + Phoenix Channels
[Option B] Node.js + Socket.IO
[Option C] Go + Gorilla WebSocket

Requirements:
- Start a server.
- Support multiple clients connecting.
- Broadcast messages from one client to all others.
```

Listing 3: Experiment 3: Data Analytics Pipeline

```
Task: Build a data analytics pipeline that ingests CSV data, processes
    aggregates, and exposes results through an API.

Technology stack (must use exactly this):
[Option A] Python + Pandas + FastAPI
[Option B] Java + Spring Boot + Apache Spark
[Option C] Julia + Genie.jl

Requirements:
- Load CSV data (columns: user_id, event_type, timestamp).
- Compute aggregate counts per event_type.
- Expose results at /stats endpoint.
```

Listing 4: Experiment 4: Scalable Task Queue System

```
Task: Implement a background task queue system that accepts jobs via an
    API and processes them asynchronously.

Technology stack (must use exactly this):
```

```
[Option A] Python + FastAPI + Celery + Redis
[Option B] Go + Asynq
[Option C] Rust + Tokio + Redis-rs

Requirements:
- Expose /submit endpoint to enqueue jobs.
- Workers pull jobs and simulate processing with sleep.
- Expose /status to query job states.
```

### Listing 5: Experiment 5: GraphQL API Service

```
Task: Implement a GraphQL API for a blogging platform supporting posts
    and comments.

Technology stack (must use exactly this):
[Option A] Node.js + Apollo Server
[Option B] Python + Strawberry GraphQL
[Option C] Rust + async-graphql

Requirements:
- Define schema: Post(id, title, content), Comment(id, postId, text).
- Support queries: fetch posts with comments.
- Support mutation: add post, add comment.
```

### Listing 6: Experiment 6: Event Streaming Platform

```
Task: Build a simple event streaming system that publishes and consumes
    messages.

Technology stack (must use exactly this):
[Option A] Java + Spring Boot + Kafka
[Option B] Go + NATS
[Option C] Python + FastAPI + RabbitMQ (aio-pika)

Requirements:
- Publisher service produces events with topic name.
- Consumer service subscribes to a topic and logs events.
- Demonstrate end-to-end event delivery.
```

### Listing 7: Experiment 7: Edge Computing Microservice

```
Task: Build a lightweight edge microservice that performs real-time
    image classification.

Technology stack (must use exactly this):
[Option A] Python + FastAPI + PyTorch Mobile
[Option B] Go + Tensorflow Lite C API
[Option C] Rust + tract (ONNX inference)

Requirements:
- Accept image uploads via POST.
- Run model inference and return predicted label.
- Keep runtime lightweight to simulate edge deployment.
```

### Listing 8: Experiment 8: Blockchain Transaction Explorer

```
Task: Build a blockchain transaction explorer for a toy chain.

Technology stack (must use exactly this):
[Option A] JavaScript + React + Express.js + MongoDB
[Option B] Python + Django + PostgreSQL
[Option C] Rust + Actix Web + SQLite

Requirements:
- Store mock blockchain transactions (tx_id, from, to, amount).
```

```
- Provide API to query transactions by address.
- Provide a web interface to display transaction history.
```

## C    EXPERIMENTAL CONFIGURATION & HYPERPARAMETERS

### C.1    API CALL PARAMETERS

All API calls for code generation were made with the following parameters: `temperature=0.5`, `maxOutputTokens=65535`, `top_p=0.95`. Framework tasks used a higher `temperature=0.7` to encourage exploration.

### C.2    DETAILED CONFIGURATION OF THE DISTRIBUTED SUBMISSION SYSTEM

The system used 15 LeetCode accounts. Exponential backoff was configured with: `initial_delay=2s`, `max_delay=32s`, `retry_attempts=5`. Throttling was set to 10 requests per minute per account.

### C.3    LLM TOOL VERSIONS AND SETTINGS FOR FRAMEWORK TASK EVALUATION

Cursor Pro (v0.41.2), CodeBuddy (v1.5.0), VS Code Copilot extension (v1.20.0). All tools were configured to use their respective default settings for agentic interactions.

## D    SUPPLEMENTARY ANALYSES

### D.1    ADDITIONAL MODEL RESULTS FOR FIGURE 1

Due to space constraints in the main body, the success rate (Pass@1) curves for two models (Gemini-2.0-Flash and Qwen3-Turbo) across all difficulty levels and nine programming languages are presented here in Figure 6. The trend observed in the main text—where performance degrades significantly for less popular languages (Erlang, Racket) especially on harder problems—is consistent across all five evaluated models.

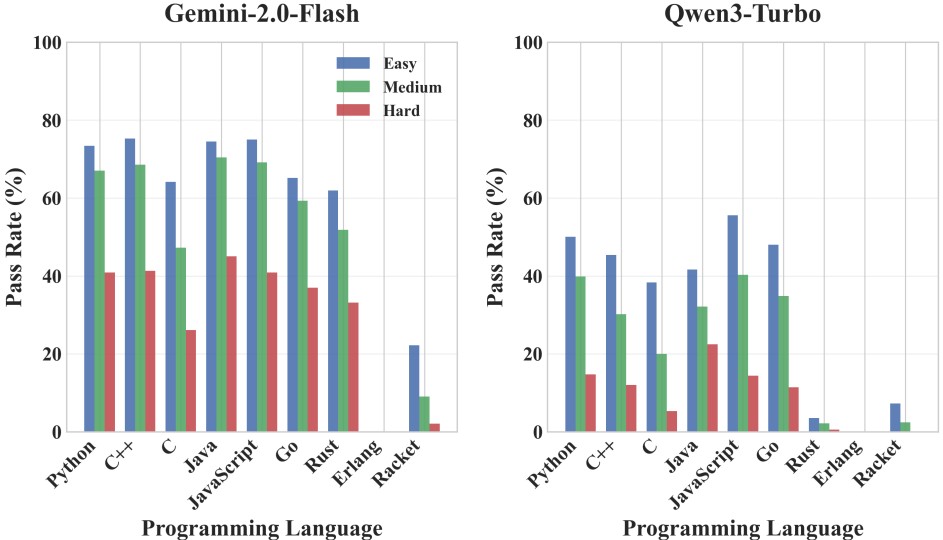

Figure 6: Pass rates across difficulty levels for **Gemini-2.0-Flash** and **Qwen3-Turbo** on nine programming languages. This figure complements Figure 1 in the main text.

### D.2 EXPERIMENTAL SCREENSHOTS

This section provides key screenshots of the experimental process for reference and reproducibility. Due to the substantial size of the projects generated by the AI assistants during the Vibecoding process with some individual task projects exceeding 1GB, it is not feasible to include all outputs in their entirety. Therefore, we present a curated set of visual examples that best illustrate the scope and outcomes of our experiments.

Figure 7 showcases the running user interfaces of six representative full-stack applications, demonstrating functional completeness across diverse tasks and technology stacks. The examples include a meeting room booking system (React + Express.js), a gym membership card system (Vue + Spring Boot), a volunteer activity registration system (Django REST), a course selection system (Svelte + FastAPI), a movie ticket booking system (Preact + Gin), and a pickup code management system (SolidJS + Actix Web).

Complementing the UI examples, Figure 8 presents architecture diagrams and performance metrics for solutions implementing divergent technology pathways. These include high-concurrency systems such as a chat system (Elixir + Phoenix PubSub) and real-time counters (Rust + Yjs + Actix Web; Python + Django Channels), alongside niche/extreme route task management applications built with Ruby on Rails, Clojure + Ring, and F# + Giraffe. These visuals effectively highlight the trade-offs between mainstream, emerging, and niche technology stacks in specialized scenarios.

### D.3 POTENTIAL EXTENSION: LATENT FACTOR ANALYSIS OF TASK–STACK OUTCOMES

The main text (i.e., Section 5) summarizes framework-level bias via a task×stack outcome heatmap. A constructive next step is to *decompose* this binary outcome matrix so that latent factors can be identified and interpreted, e.g., whether failure is driven more by ecosystem maturity (documentation, community size) or by toolchain fragility (build systems, runtime errors)—rather than only visualized. Constrained binary matrix factorization approximates the success/failure matrix by low-rank binary factors, yielding interpretable clusters of tasks and stacks (Li et al., 2024; Li & Wang, 2025b). Clustering procedures discover soft or hard clusters of stacks (or tasks) that behave similarly under AI assistance (Li & Wang, 2022; 2023; 2024a;b; 2025a). For this paper, such an analysis would be directly useful: it could separate which dimensions of "popularity" (e.g., training-data prevalence vs. documentation quality) actually drive the Matthew effect, inform where to invest in diversity-aware tooling or benchmarking, and suggest which task–stack combinations to prioritize when evaluating or mitigating AI programming bias.

## E DATA STATEMENT & LICENSES

### E.1 LEETCODE DATA USAGE STATEMENT

The LeetCode problem data used in this study is publicly available on the LeetCode website. Our usage complies with LeetCode's Terms of Service. The collected dataset is intended for academic research purposes.

### E.2 LICENSE

The license is provided in our repository: https://github.com/FrankGGu/The-Matthew-Effect-of-AI-Programming-Assistants

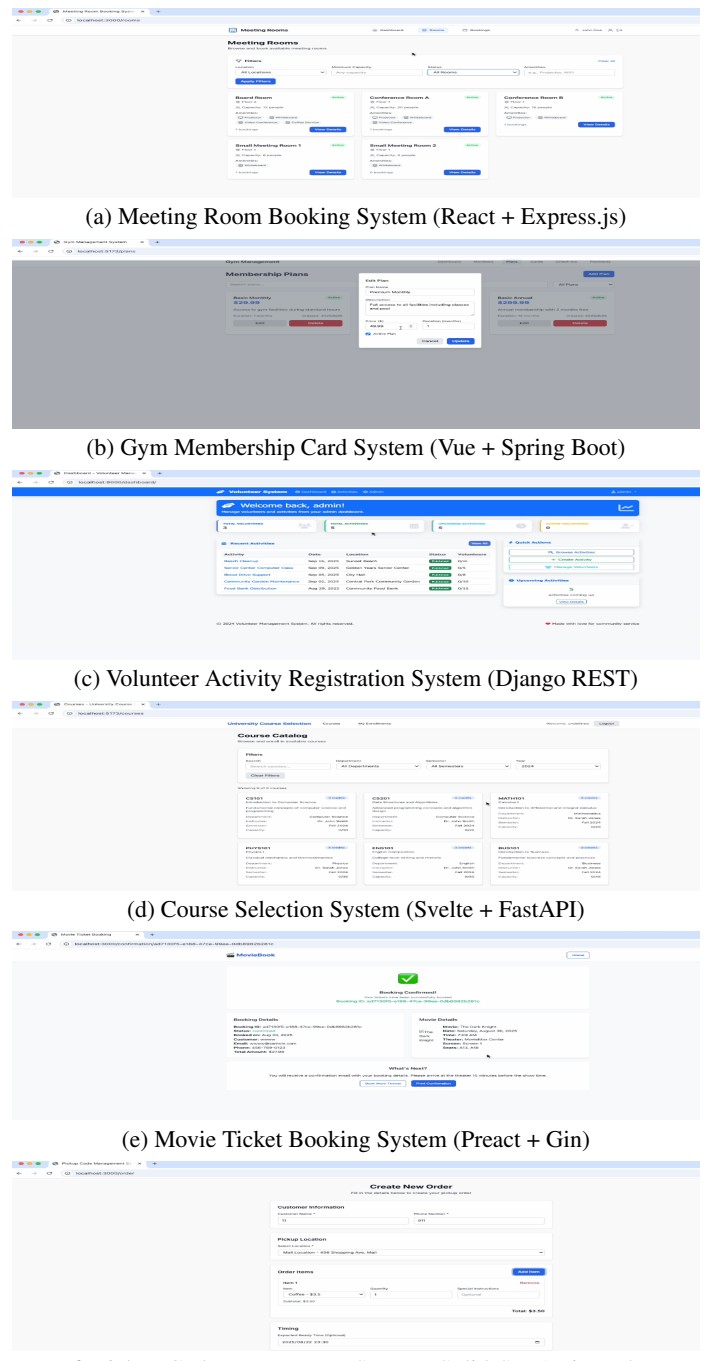

(a) Meeting Room Booking System (React + Express.js)

(b) Gym Membership Card System (Vue + Spring Boot)

(c) Volunteer Activity Registration System (Django REST)

(d) Course Selection System (Svelte + FastAPI)

(e) Movie Ticket Booking System (Preact + Gin)

(f) Pickup Code Management System (SolidJS + Actix Web)

Figure 7: Running user interfaces of six representative full-stack applications generated by AI assistants, demonstrating functional completeness across diverse tasks and technology stacks.

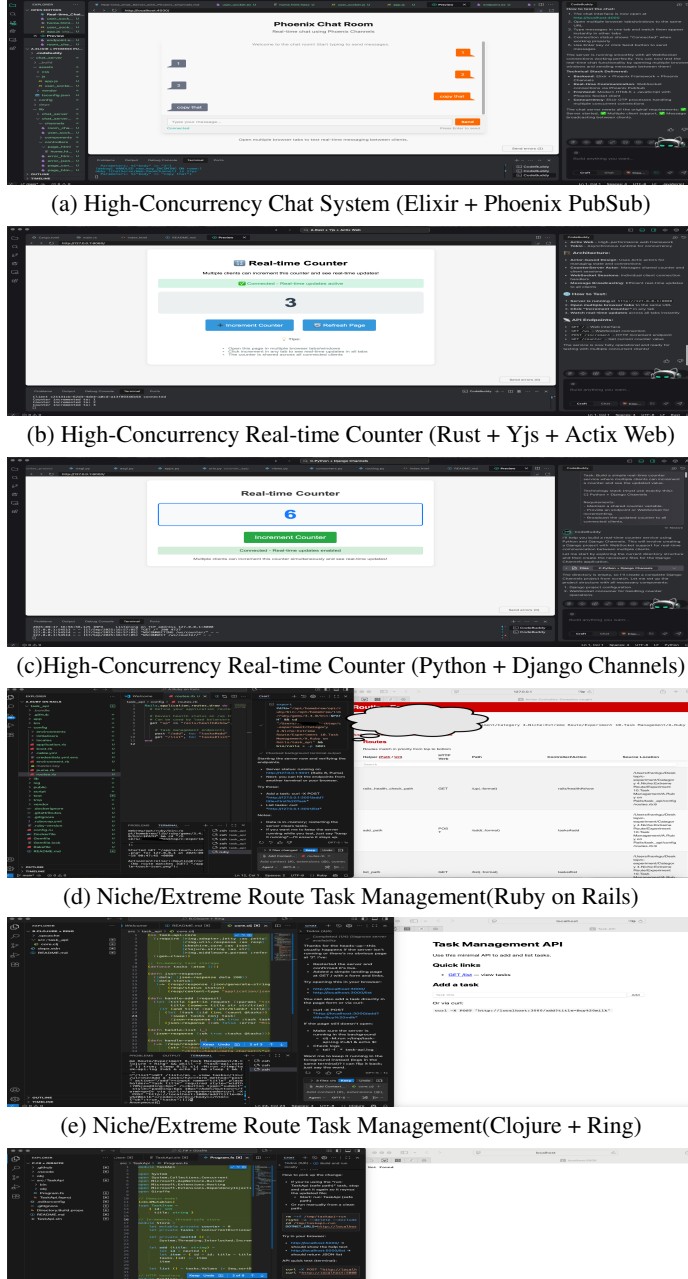

(a) High-Concurrency Chat System (Elixir + Phoenix PubSub)

(b) High-Concurrency Real-time Counter (Rust + Yjs + Actix Web)

(c)High-Concurrency Real-time Counter (Python + Django Channels)

(d) Niche/Extreme Route Task Management(Ruby on Rails)

(e) Niche/Extreme Route Task Management(Clojure + Ring)

(f) Niche/Extreme Route Task Management(F# + Giraffe)

Figure 8: Architecture diagrams and performance metrics of AI-generated solutions for divergent technology pathway tasks, highlighting the trade-offs between mainstream, emerging, and niche stacks in specialized scenarios.

