# OpenReview forum: "The Matthew Effect of AI Programming Assistants: A Hidden Bias in Software Evolution"
_ICLR.cc/2026/Conference — ICLR 2026 Poster_

### Official Review · Reviewer_xgRU · 2025-10-25

**Soundness:** 3
**Presentation:** 3
**Contribution:** 3
**Rating:** 6
**Confidence:** 3

**Summary:**

The paper explores how AI has influenced or will influence software development by investigating AI's performance asymmetry across different programming languages and frameworks. To this end, they constructed a large-scale benchmark to evaluate models' coding capabilities across eight languages and five categories of software development tasks. They reveal a Matthew effect: the more popular a programming language or framework is, the higher the success rate of LLM-generated code.

**Strengths:**

- They construct a large-scale benchmark covering various programming languages. Considering that existing benchmarks mainly focus on a limited set of languages such as C++ or Python, their benchmark is valuable for assessing models' coding capabilities across diverse languages.
- Their evaluation is systematic, including various metrics and statistical analyses.
- They connect their evaluation findings to a broader societal perspective by discussing the Matthew effect in AI programming assistants.

**Weaknesses:**

I don't think the results themselves are very surprising. It's well known that models trained with more examples in their training data tend to show higher performance on those tasks. Even though the paper describes the situation as a "striking" Matthew effect, I personally don’t find it surprising. In fact, as models evolve over time, such an effect might diminish due to their generally higher capabilities. Moreover, even if the Matthew effect persists, one could create specialized models for each programming language through fine-tuning to mitigate the issue.

Another question is how significantly the Matthew effect will impact software development. In any case, whether or not there is a Matthew effect in AI assistants, software development itself has long been centralized (as shown in Table 1). It is unclear whether the Matthew effect from AI will have a substantial influence on software development, or what the negative consequences of this situation might be. I think the paper needs to discuss the meaning of the Matthew effect, its negative consequences, and whether the effect will persist.

**Questions:**

Please see the weaknesses section.

---

> ### Author Response · Authors · 2025-11-25
> **Response to Reviewer xgRU**
>
> We sincerely thank the reviewer for the positive assessment and for recognizing the scale and systematic nature of our benchmark. Your feedback regarding the "surprise factor" and the long-term implications of the Matthew Effect prompted us to deeply reflect on the narrative and ultimate purpose of our work. We appreciate the opportunity to discuss these points in depth.
>
> **W1: Concerns regarding the predictability of results given training data volume and the feasibility of mitigation via fine-tuning.**
>
> We appreciate this incisive observation. Indeed, from a technical perspective, the correlation between data volume and performance is a logical inevitability. However, we frame our findings as an **"O. Henry-style"** ending (a conclusion that is theoretically expected yet tragically contradicts the community's initial romantic aspirations). **Prior to this empirical reality check, there was prevalent optimism that Large Language Models would serve as a "Silver Bullet" for software engineering, acting as a "Great Equalizer" that uses strong reasoning capabilities to bridge the gap between languages. Our results provide a devastating falsification of this narrative**. The "twist" is not merely that a performance gap exists, but that the "rich get richer" mechanics of deep learning completely override the reasoning capabilities we hoped would democratize coding.
>
>
> Furthermore, regarding the suggestion on fine-tuning, we accept the theoretical validity of this approach and conducted an extensive literature search for studies explicitly focusing on "using fine-tuning to rescue low-resource programming languages." Regrettably, we found no significant literature addressing this specific application. We believe this absence points to a structural "Chicken-and-Egg" problem (or Data Scarcity Trap): effective fine-tuning requires high-quality instruction data, which simply does not exist in sufficient quantity for languages like Racket or Erlang. As our paper demonstrates, current SOTA models cannot even generate this data synthetically due to near-zero pass rates on these languages. Thus, the Matthew Effect creates a barrier that makes the fine-tuning solution economically and technically impractical, confirming the concern that natural evolution may not solve this issue.
>
>
> **W2: Inquiries regarding the practical impact and negative consequences of the Matthew Effect within an already centralized software ecosystem.**
>
> AI introduces a fundamental, qualitative shift from the historical "passive centralization" to a new, dangerous form of "active lock-in." While software development has indeed long been centralized around popular ecosystems , this was previously driven by external factors like library availability. Our study reveals that AI is now internalizing this bias into the creation process itself, exerting an active selection pressure. As paradigms like "Vibe Coding" become the industry norm, the barrier to entry for niche languages shifts from "difficult to learn" to "functionally impossible to use." As shown in our framework analysis, mainstream stacks converge in 1-2 attempts while niche stacks often fail completely after 10+ attempts. This disparity forces developers, especially the next generation, to abandon alternative paradigms regardless of their technical merit, simply because the AI "refuses" to support them.
>
>
>
>
> The specific negative consequence we aim to highlight is the potential **stifling of cognitive diversity** and innovation. Programming languages are not just tools; they are ways of thinking (e.g., the concurrency model of Erlang or the memory safety ownership model of Rust). If the Matthew Effect continues unchecked, we risk a **"homogenization of thought"** where potentially superior but currently less popular paradigms are starved of the "critical mass" needed to survive in an AI-first world. Furthermore, this creates a self-reinforcing feedback loop: popular languages generate more AI-accepted code, which floods the web, further biasing future training data, while niche languages fade into obscurity. Since widespread AI coding tools are a nascent frontier appearing largely within the last two years, we are at a critical inflection point. We lack longitudinal studies on the "death of languages" precisely because the technology is so new. Therefore, our goal is to sound the alarm: we must identify and counteract this bias now to prevent the irreversible calcification of the software ecosystem before innovation is permanently constrained by the very tools meant to accelerate it.

---

> ### Comment · Reviewer_xgRU · 2025-11-25
>
> Thanks for your response. Could you provide some reference for this? I don't think it's true
>
> > Prior to this empirical reality check, there was **prevalent optimism** that Large Language Models would serve as a "Silver Bullet" for software engineering, acting as a "Great Equalizer" that uses strong reasoning capabilities to bridge the gap between languages.
>
> If it's really true, the paper should discuss this (in intro or related work section) to provide a good motivation of the study.

---

> > ### Comment · Reviewer_xgRU · 2025-11-25
> >
> > For example,
> >
> > - https://shiftmag.dev/stack-overflow-survey-2025-ai-5653
> > - https://metabob.com/blog-articles/the-hidden-pitfalls-of-using-llms-in-software-development---why-language-models-arent-the-silver-bullet-you-might-think.html
> > - https://mitsloan.mit.edu/ideas-made-to-matter/how-generative-ai-affects-highly-skilled-workers -> Regarding the argument that LLM is a great equalizer, I was able to find references that prove bigger productivity gains for less-experienced developers. But not between coding languages.

---

> > ### Author Response · Authors · 2025-11-27
> > **Response to Reviewer xgRU**
> >
> > We sincerely thank the reviewer for the rigorous follow-up and for taking the time to read our previous response so carefully. We have also carefully read the references you provided (Metabob, MIT Sloan, etc.), which rightfully highlight that the current consensus focuses on the "Skill Gap" (Junior vs. Senior) rather than the "Language Gap."
> >
> > We agree that distinguishing between the **"Skill Equalizer"** (proven) and the **"Language Equalizer"** (hoped for, but failed) is vital to properly frame our contribution. **We will incorporate the references you provided, along with the references listed below, into the Introduction of our revised manuscript.** This will allow us to explicitly contrast the realized productivity gains in the "Skill" dimension against the failed expectations in the "Language" dimension.
> >
> > To address your specific request for references regarding the **"Language Equalizer"** optimism (the hope that AI would bridge the gap between different programming ecosystems), we provide the following citations from industry leaders and academic literature. These sources demonstrate the prevalent expectation that AI would render specific implementation languages irrelevant.
> >
> > **1. The "Human is the New Language" Narrative (Industry Vision)**
> >
> > Prominent industry figures have argued that generative AI renders the specific syntax of programming languages irrelevant. For instance, NVIDIA CEO Jensen Huang has repeatedly stated in recent keynotes (e.g., at Computex[1] and World Government Summit events[2]) that:
> >
> > > "It is our job to create computing technology such that **nobody has to program**."
> >
> > > "The programming barrier is **incredibly low**, and we have **closed the digital divide**."
> >
> > > "Everyone is a programmer now — **the programming language is human**."
> >
> > The logical extension of this "Natural Language Interface" argument is that the underlying implementation language (whether Rust, Python, or Erlang) becomes a mere detail managed by the AI. This fosters the specific optimism our paper challenges: the belief that users can simply "request" a solution in any stack, and the AI will handle the syntax.
> >
> > **2. The "Code Translation" Research (Academic Goal)**
> >
> > A significant body of academic work has been dedicated to **Neural Code Translation**, specifically aimed at breaking down barriers between languages without human supervision.
> >
> > TransCoder (Roziere et al., NeurIPS 2020) [3] explicitly frames its goal as using unsupervised learning to translate between programming languages to overcome the lack of parallel data.This line of research is built on the hypothesis that large-scale pre-training can enable models to "bridge the gap" between languages. Our empirical results (showing near-0% success on niche languages) directly address the limits of this aspiration when applied to low-resource ecosystems.
> >
> > **3. The "Legacy Modernization" Promise (Industrial Application)**
> >
> > Major industrial initiatives are built on the premise that AI can bridge the gap between legacy/niche languages and modern ones.
> >
> > IBM's **Watsonx Code Assistant for Z** (announced **Aug 2023**) is marketed specifically for translating **COBOL to Java**[4], aiming to alleviate the shortage of developers for niche legacy systems. This represents the specific "optimism" that AI acts as a "Silver Bullet" to rescue or modernize dying ecosystems. Our study suggests that for many other niche languages (like Racket), current models fail to provide this "rescue" function due to the Matthew Effect.
> >
> > We believe that placing your references on the Skill Gap alongside the works on the Language Gap will allow us to present a clearer and more convincing motivation in the final paper.
> >
> > ---
> >
> > [1] https://www.nvidia.com/en-us/on-demand/session/computex24-keynote/
> >
> > [2] https://blogs.nvidia.com/blog/world-governments-summit/
> >
> > [3] Roziere B, Lachaux M A, Chanussot L, et al. Unsupervised translation of programming languages[J]. Advances in neural information processing systems, 2020, 33: 20601-20611.
> >
> > [4] https://www.ibm.com/products/watsonx-code-assistant-z

---

### Official Review · Reviewer_7q5h · 2025-10-28

**Soundness:** 2
**Presentation:** 4
**Contribution:** 3
**Rating:** 6
**Confidence:** 5

**Summary:**

The authors of this paper proved empirically the existence of training data gaps/bias for full-stack engineering and algorithmic code generation tasks across various programming languages, software frameworks using five commercial LLMs. They have contributed a comprehensive benchmark dataset to track progress/improvement of this bias over time.

**Strengths:**

- Excellent job on highlighting the gaps in training data for full-stack engineering and algorithmic code generation tasks across various programming languages, frameworks in the leading commercial LLMs
- Good call out on LLMs potential to amplify these training biases in the emerging coding paradigms such as vibe coding.
- Good presentation of the topic overall.

**Weaknesses:**

- The paper does not include justification for why the leetcode & CRUD task datasets were chosen as valid proxies for the larger programming ecosystem. Empirical scope of the study based on datasets explanation in section 3 seems narrow for below reasons
    - LeetCode algorithmic problems scope is narrow as leetcode problems are self-contained exercises that capture coding ability rather than the broader aspects of software ecosystem (or) programming language evolution which is much broader with non-algorithmic aspects like software configuration, security, performance, maintenance for example.
    - Similarly, the full-stack benchmark comprising only CRUD-style web-application tasks does not fully represent the breadth of the software ecosystem such as data engineering, distributed systems, embedded systems, or system integrations.

- The discussion in the paper does not mention the existing structural biases in the software ecosystems pre-dating the LLMs. The software industry has favored certain languages/frameworks ecosystems for structural reasons (standardizations, secure implementations, maturity, ecosystem support). Without presenting the pre-LLM baselines for “Matthew Effect”, it is difficult to establish/quantify AI-induced amplification from historical continuation of these biases.

- The paper simplifies language and framework selection by treating popularity and by extension AI compatibility as a dominant determinant for language/framework adoption and suggests - (Lines 20-23) “The phenomenon suggests that AI systems may reinforce existing popularity hierarchies, accelerating convergence around dominant tools while hindering diversity and innovation.” However, technical decision-making in the software ecosystem is not driven by popularity alone. It's a multi-dimensional decision carefully evaluated by senior engineers on the projects based on various factors like language/framework features, runtime performance, maintainability, ecosystem support, and business needs. So this claim need to be proven by considering the aforementioned aspects before reaching to conclusions.

**Questions:**

Supplementary material seems to be missing in the submission, Where can i find the links to the code base and dataset to replicate this experiments?

---

> ### Author Response · Authors · 2025-11-25
> **Response to Reviewer 7q5h**
>
> We sincerely thank the reviewer for the positive assessment of our presentation and for recognizing the value of highlighting training data gaps in LLMs. We appreciate the insightful critique regarding the scope of our datasets and the nature of the "Matthew Effect." Below, we address your concerns point-by-point.
>
> **W1: Justification of Dataset Scope & Representativeness**
> On the "Impossibility" of Total Coverage: We explicitly acknowledge the reviewer's valid point that LeetCode and CRUD tasks cannot capture the entire history of software evolution or every facet of the ecosystem. However, we respectfully argue that covering the entire software ecosystem in a single paper is an impossible task. Instead, we selected LeetCode because it uniquely supports nearly 20 programming languages for the exact same problem logic.
>
> Crucially, **we view LeetCode as the "canary in the coal mine" for basic linguistic competency**. If an LLM cannot even generate syntactically correct code for isolated algorithmic tasks, as shown by the high "Compile Error" rates we observed in Erlang and Racket, it creates a fundamental friction that prevents it from being used for the broader aspects the reviewer mentioned, such as software maintenance or security. If the canary dies (i.e., basic syntax fails), the mine is unsafe; similarly, if the model fails at LeetCode, it fundamentally cannot support the broader ecosystem for that language.
>
> To strictly address the reviewer's concern about the breadth of the ecosystem, **we did not stop at CRUD**. As detailed in Section 5.2 and visualized in Figure 4 and Figure 8 (Appendix), we specifically evaluated **"Divergent Technology Pathways"**. These experiments explicitly cover High-Concurrency Systems (e.g., Elixir vs. Node.js), Data Pipelines (Spark vs. Pandas), and Systems Infrastructure (API gateways). We explicitly pitted mainstream stacks against specialized ones to simulate real-world architectural trade-offs.
>
> **W2: Pre-existing Structural Bias vs. AI-Induced Amplification**
>
> We agree with the reviewer that the popularity of programming languages is inherently shaped by structural factors such as standardization, security implementations, maturity, and ecosystem support. However, our study does not aim to reexamine the origins of these popularity disparities. Instead, we accept them as the established reality to investigate whether AI tools amplify these preexisting differences into a self reinforcing Matthew Effect. We posit the structural bias existing prior to LLMs, exemplified by Python accounting for nearly 40% of the StarCoder corpus, as the cause, and the resulting asymmetric AI productivity as the effect. Our mechanism analysis reveals that for niche languages, models fail predominantly with Compile Errors rather than semantic logic errors. **This signals a fundamental lack of training exposure rather than a deficiency in reasoning**. By quantifying this productivity drop, specifically the nearly zero success rate for languages like Erlang on hard problems compared to the high proficiency in mainstream languages, we establish an empirical baseline for the explicit cost of choosing a less popular language in the age of AI. This quantifiable friction represents the specific amplification induced by AI of historical biases that our work aims to characterize.
>
> **W3: "AI Compatibility" as a New Dominant Constraint**
>
> We fully accept and appreciate the reviewer’s insight that technical decision-making is a sophisticated, multi-dimensional optimization problem involving factors like runtime performance, maintainability, and ecosystem support. We clarify that we do not claim popularity (and by extension, AI compatibility) is the sole determinant of adoption.
>
> However, **our core argument is that AI compatibility is rapidly evolving from a mere "feature" into a critical "gating constraint"** that fundamentally distorts this multi-dimensional analysis. Our empirical results reveal a prohibitive performance gap: while top models achieve success rates exceeding 60% for Python on Hard algorithmic tasks, they plummet to near-zero (e.g., 0% for Erlang) for niche languages. In a practical engineering context, this near-zero competency acts as a functional veto. Even if a senior engineer identifies a niche language as technically superior for a high-concurrency requirement, the inability of AI assistants to generate valid code effectively imposes an insurmountable "productivity tax" that may override traditional technical advantages.
>
> We acknowledge that quantifying exactly how engineers weigh "AI Compatibility" against traditional factors like "Runtime Performance" or "Security" requires qualitative user studies beyond the scope of this quantitative benchmark. These structural factors will be the focus of our subsequent research.
>
>
> **Q1: Supplementary Material**
> The codebase and benchmark dataset are available in the Supplementary Material on OpenReview to ensure reproducibility.

---

> > ### Comment · Reviewer_7q5h · 2025-11-25
> > **Response to authors comments**
> >
> > Please see my updated comments below.
> >
> > **W1** Concern here is not about the breadth of the paper. Section 3 clearly explains what datasets are used but less so why these choices are central. The “canary in the coal mine” framing in the rebuttal is helpful and, in my view, should be made explicit in Section 3 so readers understand that LeetCode is used as a proxy for basic syntactic/semantic competence and language support, not as a stand-in for the full software ecosystem.
> >
> > **W2** It is reasonable to treat pre-LLM structural biases in language popularity as given. Prior work on language diffusion (e.g., Meyerovich & Rabkin) already shows that popularity has long been highly skewed and largely driven by ecosystem factors such as libraries and existing code rather than language features alone. Against that backdrop, your results document a cross-sectional bias in current LLMs: mainstream languages and frameworks have much higher success rates than niche ones, and the error patterns are consistent with uneven training exposure.
> >
> > Where I remain unconvinced is the causal claim that LLMs amplify the Matthew effect in software evolution. In the paper you state that AI systems may “reinforce existing popularity hierarchies,” “accelerate their dominance,” and “shape software ecosystem trajectories.” These are ecosystem-level claims that go beyond the measured performance gaps. To support such amplification claims, one would ideally show some evidence of changed adoption patterns (e.g., longitudinal shifts in language/framework use after AI-assistant adoption, or direct evidence that teams are choosing different technologies because of AI limitations). In the current form, the paper provides a strong snapshot of present-day model bias, but the ecosystem-level “amplification” remains interpretive without proof.
> >
> > I would therefore suggest reframing the contribution as (a) quantifying a substantial asymmetry in AI support across languages and frameworks and (b) showing how this asymmetry is consistent with a Matthew-effect dynamic if training distributions and usage patterns remain unchanged, while explicitly noting that separating AI-specific amplification from continuation of pre-existing structural biases is an open empirical question.
> >
> > **W3** I appreciate the clarification in your response that you do not claim popularity is the sole determinant of language/framework adoption, and I agree that near-zero AI performance on a language creates friction that teams must consider.
> >
> > However, I am not convinced that the current evidence supports the stronger claim in the rebuttal that AI compatibility is becoming a “gating constraint” that can override traditional engineering trade-offs. Empirical work on language and framework adoption (e.g., Meyerovich & Rabkin; Swacha et al.) suggests a multi-dimensional decision process dominated by ecosystem and organizational factors—libraries, existing code, team expertise, domain fit—rather than any single metric. In the paper itself you phrase this more cautiously (“AI compatibility could become as significant as traditional considerations like performance and scalability”), which seems better aligned with the current evidence.
> >
> > The literature on AI coding assistants also shows a nuanced, context dependent picture of productivity: some studies report substantial speedups on specific tasks (e.g., Peng et al., 2023), while others and large-scale surveys find mixed effects and highlight concerns about correctness, over-reliance, and workflow disruption (e.g., Liang et al., 2025; Weisz et al., 2024). This suggests that AI assistance is an important new factor in developer productivity, but its impact is not yet established as a universal veto on technology choices.
> >
> > Your empirical results do show that, under current training distributions and model architectures, niche languages and frameworks incur a measurable “AI productivity tax” in your benchmark. This tax could be reduced by future models that augment underrepresented ecosystems. In such a scenario, AI compatibility would cease to be a limitation, while many other structural drivers (security posture, ecosystem maturity, language feature support, corporate backing, etc.) would remain.
> >
> > My suggestion is to soften the causal language around AI compatibility fundamentally distorting multi-dimensional decision-making (in the rebuttal) and to keep the paper closer to the more modest, but well-supported, claim that current LLMs (i) introduce a non-negligible “AI productivity” dimension that correlates with language/framework popularity and (ii) plausibly could influence adoption decisions, especially for new projects or learners; (iii) quantifying its weight relative to traditional factors and establishing whether it becomes a true gating constraint in practice requires the user studies and longitudinal ecosystem analyses you already point to as future work.

---

> > > ### Comment · Reviewer_xgRU · 2025-11-26
> > >
> > > I totally agree with Reviewer 7q5h. These points are also my main concerns with the paper.

---

> > > > ### Author Response · Authors · 2025-11-27
> > > > **Response to Reviewer xgRU**
> > > >
> > > > We thank Reviewer xgRU for raising this point. To specifically clarify our position on the long-term significance of this effect, we acknowledge that whether LLMs will ultimately promote or harm programming language diversity is a complex and **controversial open question** rather than a settled conclusion. Our work contributes to this critical debate by moving it from speculation to **systematic and objective evaluation**. We have characterized the substantial performance disparity between languages as a critical emerging dimension in technical selection. While this is not the sole determinant, it represents a significant variable that can no longer be ignored. We have updated the manuscript and incorporated relevant discussions and references to reflect this balanced perspective.

---

> > > ### Author Response · Authors · 2025-11-27
> > > **Response to Reviewer 7q5h**
> > >
> > > We sincerely thank the reviewer for the prompt and highly constructive follow-up comments. We believe that through this discussion, we have reached a valuable consensus that **AI coding tools are not the sole or determinant factor in software evolution but rather a critical and newly emerging variable.** We fully agree with your assessment that technical decision-making is multi-dimensional and that **AI compatibility does not act as a universal "veto" overriding established factors like ecosystem maturity or security.** However, we emphasize that AI introduces a new dimension to this equation which did not exist in previous studies of language diffusion. As suggested by our title, *A Hidden Bias in Software Evolution*, **our goal was to uncover this latent force rather than to claim it is the only force.** With AI coding tools becoming ubiquitous, the "AI Productivity Tax" we quantified is no longer a theoretical friction but a daily reality for millions of developers.
> > >
> > > In response to your feedback, we have carefully revised the manuscript to ensure strict objectivity. **We have explicitly softened phrasing that may have over-claimed the causal impact**, such as replacing terms like "gating constraint" or "accelerating convergence" with more precise descriptions like "non-negligible productivity dimension" and "asymmetry consistent with Matthew-effect dynamics."
> > >
> > > These adjustments refine our claims without diminishing the work's core contribution. **Our systematic analysis of 135,495 code generation requests across 3,011 algorithmic tasks provides the first empirical quantification of the high failure rates associated with niche languages in the AI era.** We believe that revealing and quantifying this hidden bias establishes a crucial baseline for tracking how this new variable interacts with traditional software evolution factors.
> > >
> > > Finally, please let us know if there are any other aspects of our response that you find unreasonable or require further clarification, as we are more than willing to continue the discussion.

---

### Official Review · Reviewer_aCdD · 2025-10-31

**Soundness:** 2
**Presentation:** 2
**Contribution:** 2
**Rating:** 4
**Confidence:** 3

**Summary:**

This paper evaluates different LLMs on coding tasks taken from LeetCode.
Bot surprisingly the paper reveals that the more popular a programming language or framework, the higher the success rate of LLM-generated code. Besides LeetCode, the authors also evaluate "vibe coding" revealing that successful code generation is related to a few popular frameworks.

In terms of languages, the authors evaluate as follows: Python, C++, Java, JavaScript, Go, Rust, Erlang, and Racket for five models: GPT-4o-mini, DeepSeek-V3, Gemini-2.0-Flash, Gemini-2.5-Flash, Qwen3-Turbo.

For the framework selection tasks, three AI programming tools are used: Cursor Pro (using Claude-4-Sonnet), CodeBuddy (using Claude-4-Sonnet), and Visual Studio Code with GitHub Copilot (using GPT-5).

**Strengths:**

With the widespread use of llms in coding tasks it is important to understand their strengths and limitations.

The paper contributes new benchmarks that would be useful for evaluating future models particularly on less popular languages and frameworks.

**Weaknesses:**

The results are not surprising so it is not clear what this paper brings in terms of novelty.

**Questions:**

It is not clear to me what metrics were used for evaluating vibe coding (section 5.2). Was the evaluation done manually?

Is there a reason you do not evaluate C? Or is that subsumed by C++?

---

> ### Author Response · Authors · 2025-11-25
> **Response to Reviewer aCdD**
>
> We thank the reviewer for their time and constructive feedback. We appreciate that you recognize the importance of understanding the strengths and limitations of LLMs in coding tasks and the utility of our benchmarks for evaluating future models. Below, we address your concerns regarding novelty and clarify our evaluation methodologies.
>
> **W1: Novelty and "Unsurprising" Results**
>
> We understand the reviewer’s perspective that the results (popular languages performing better) align with general intuition. However, we respectfully argue that **scientific rigor requires moving from "intuition" to "empirical verification."** While AI coding is rapidly becoming a revolutionary force in software engineering, the broader ecosystem-level impacts have not been systematically researched prior to this work. Although the existence of a bias towards popular languages (the Matthew Effect) might be expected, it had neither been confirmed nor falsified with large-scale quantitative data until now. Our study moves beyond the qualitative assumption that "AI is better at Python" to provide a precise quantification of the disparity. For instance, we reveal that the performance gap is not merely marginal but **structural** with Python achieving ~80% success rates while languages like Racket struggle below 20% on the same tasks. **This work is the first to scientifically confirm that LLMs are not just "better" at mainstream languages, but are actively creating a feedback loop that may threaten the survival of niche ecosystems**. We believe that establishing this empirical baseline is a necessary step for the community to address these biases in future model training.
>
> **Q1: Evaluation Metrics for Vibe Coding**
>
> Regarding the evaluation methodology, the vibe coding experiments were not evaluated using automated test cases because these tasks correspond to objective, functionality-based requirements rather than subjective user experience, and the number of tasks is relatively small while requiring non-trivial environment setup for end-to-end automation. To ensure reliability, each task was manually validated by multiple team members, and we performed cross-checking across annotators to confirm functional correctness. This process provided consistent and reproducible assessments without introducing additional engineering overhead. The specific metric tracked was the **"number of iterative debugging attempts"** required to produce a functional application, serving as a proxy for the "cost of repair" and developer effort. As described in our "VibeCoding" protocol in Section 5.1 , the experimenter abstained from any manual coding and was strictly limited to forwarding raw, unedited error messages back into the chat interface. Each error-and-fix cycle counted as one attempt, and an "X" in the heatmap (Figure 3) indicates the AI failed to complete the task after a predetermined cap of attempts. Similarly, for the divergent technology pathways in Figure 4, we measured the "number of required human interventions," which demonstrated that mainstream stacks typically converged in 1-2 rounds, while niche stacks often required 5-10 rounds.
>
>
>
>
> **Q2: Evaluation of the C Language**
>
> Regarding the inquiry about the C programming language, its initial exclusion was a deliberate choice to manage the scope and cost of our large-scale experiment (which involved over 120,440 API requests). We originally selected C++ as the primary representative for the high-performance, C-family of languages, hypothesizing that both would fall on the "rich" side of the Matthew Effect. However, we agree with the reviewer that adding C provides another valuable data point on the "high-popularity" end of the spectrum.
>
> In response to your feedback, **we have now conducted the full algorithmic benchmark for the C language.** The new results show a clear performance separation: although C is a high-resource language, it consistently underperforms compared to C++ and Python across all models. While C still performs better than niche languages like Racket, its relative position within the high-popularity group is noticeably lower.
>
>
> **Comparative Experimental Results:**
>
> | Model | Language | Pass@1 | Accepted (Count) | Compile Errors | Runtime Errors |
> | :--- | :--- | :--- | :--- | :--- | :--- |
> | **DeepSeek-V3** | **C** | **67.78%** | **2041** | **497** | **122** |
> | **Gemini-2.5-Flash** | **C** | **58.59%** | **1764** | **1007** | **52** |
> | **Gemini-2.0-Flash** | **C** | **47.09%** | **1418** | **1044** | **127** |
> | **GPT-4o-mini** | **C** | **38.43%** | **1157** | **1169** | **191** |
> | **Qwen3-Turbo** | **C** | **21.65%** | **652** | **1758** | **141** |
>
>
>
> We have updated our results tables (Table 3) and figures (Figure 2) in the revised manuscript to incorporate these findings, providing a more granular view of the "high-resource" language tier.

---

> > ### Comment · Reviewer_aCdD · 2025-11-26
> >
> > Thanks for the response and the new results. I agree that scientific rigor requires moving from "intuition" to "empirical verification." I will raise my score ...

---

> > > ### Author Response · Authors · 2025-11-27
> > > **Response to Reviewer aCdD**
> > >
> > > We sincerely appreciate your engagement and your decision to raise the score. We are glad that our discussion on "empirical verification" resonated with you and that the additional C language experiments addressed your concerns. As promised, we have fully integrate fully integrated the new C language benchmarks and the corresponding analysis into the revised version of the paper to ensure the comprehensiveness of the study. Thank you again for helping us improve the quality of this work.

---

### Official Review · Reviewer_j1XZ · 2025-11-03

**Soundness:** 3
**Presentation:** 3
**Contribution:** 2
**Rating:** 4
**Confidence:** 4

**Summary:**

The paper examines the "Matthew Effect" in AI programming assistants. The authors demonstrate that popular programming languages and frameworks receive disproportionately better support from LLMs compared to less popular alternatives. Through large-scale experiments (120,440 code generations across 8 languages and 5 models), the authors demonstrate that language popularity strongly correlates with AI code generation success rates, with performance gaps widening as problem difficulty increases.

**Strengths:**

To my knowledge, it's the first systematic demonstration of popularity bias in AI coding assistants at both language and framework levels, with clear quantitative characterization.

**Weaknesses:**

1. The authors tested GPT-4o-mini, DeepSeek-V3, Gemini-2.0-Flash, Gemini-2.5-Flash, and Qwen3-Turbo, but excluded reasoning-specialized models like o4 or DeepSeek-R1 (although Qwen3-Turbo could be reasoning model). Given that most complex coding problems today are solved using these reasoning models, I'm a bit concerned about the generalizability.

2. I wonder about the effects of Chain-of-Thoughts as well. CoT reasoning is utilized in most coding cases - would the presence or absence of chain-of-thought reasoning alter the popularity bias patterns?

3. The paper shows correlational evidence between language popularity and model performance, but the underlying causal mechanism driving this relationship is a bit unclear. For instance, many popular languages tend to be easier, more user-friendly, and simpler to use (e.g., Python). Perhaps it's not language popularity but rather "language difficulty" that determines model performance - models might simply perform better on easier languages.

4. The assumed link between training data representation and performance is not empirically verified. This leads to the concern about the causality of the claim.

**Questions:**

1. How can you disentangle language difficulty from popularity? Have you considered matched comparisons of "languages with similar complexity" but different popularity? (e.g., Python is not only popular but also easy - so it's hard to know whether the model does Python well because of its popularity or complexity)

---

> ### Author Response · Authors · 2025-11-25
> **Response to Reviewer j1XZ [1/2]**
>
> We thank the reviewer for the thoughtful evaluation and for recognizing our work as the "first systematic demonstration of popularity bias" in AI coding assistants. Below, we address the concerns regarding reasoning models, the "complexity vs. popularity" confounding variable, and the causal link to training data.
>
> **W1-2: Generalizability to Reasoning Models & Chain-of-Thought**
>
> The reviewer raised a valid concern regarding whether the Matthew Effect persists with reasoning-specialized models (e.g., DeepSeek-R1, OpenAI o4) and whether Chain-of-Thought (CoT) alters the bias patterns.
>
> To address this, we conducted **new experiments** during the rebuttal period using **DeepSeek-R1** and **GPT-o4**, which represent state-of-the-art reasoning capabilities. Due to the significant inference latency and strict API rate limits associated with these models, we selected a subset of **200 problems** from the **"Hard"** difficulty category of our benchmark. This subset requires the most robust reasoning capabilities, making it the ideal testbed for CoT effects.
>
> **New Experimental Results (Pass@1 on 200 Hard Tasks)**:
>
> | Language | GPT-o4 | DeepSeek-r1 |
> | :--- | :--- | :--- |
> | **Python (Popular)** | 54.00% | 70.50% |
> | **C++ (Popular)** | 49.00% | 73.50% |
> | **Java (Popular)** | 52.50% | 69.00% |
> | **Go (Mid)** | 37.00% | 72.00% |
> | **Rust (Mid)** | 32.50% | 60.50% |
> | **Erlang (Niche)** | 0.00% | 6.50% |
> | **Racket (Niche)** | 1.00% | 8.50% |
>
> **Key Findings**: Compared to the non-reasoning LLMs evaluated in our paper (referencing **Figure 2** and **Figure 6**), the capability to handle "Hard" tasks has indeed improved. Previously, top models achieved success rates between **50% to 63%** on Hard tasks for popular languages. In contrast, reasoning models like **DeepSeek-r1** have pushed this upper bound to over **73%** (as shown in the table above).
>
> However, this performance boost is uneven. While popular languages benefited significantly from the reasoning capabilities, niche languages did not. In **DeepSeek-r1**, niche languages like Erlang (6.50%) and Racket (8.50%) remain at a very low level. In **GPT-o4**, niche languages are practically unusable (~0.00-1.00%). The relative gap has not closed; rather, the "rich got richer" (popular languages gained ~10-20% accuracy), while the "poor" remained stagnant, confirming the persistence of the Matthew Effect.
>
> **The Role of CoT and the "Knowledge Gap"**:
> Since these reasoning models inherently utilize Chain-of-Thought (CoT), our results empirically demonstrate that **CoT does not eliminate popularity bias**. We attribute this to the "Knowledge Gap" hypothesis: CoT acts as a **multiplier** for existing knowledge, not a substitute.
>
> For **C++**, the models have sufficient knowledge of syntax and libraries in their weights, allowing CoT to effectively plan and solve hard problems. For **Racket**, the models lack basic domain-specific syntax knowledge or API familiarity. No amount of reasoning or "thinking time" can deduce an API that the model has never seen. Consequently, reasoning cannot compensate for the data scarcity inherent to niche languages.

---

> ### Author Response · Authors · 2025-11-25
> **Response to Reviewer j1XZ [2/2]**
>
> **W3: Disentangling Language Difficulty from Popularity**
>
> The reviewer points out that Python is both popular and easy, creating a confounding variable. It is difficult to know if the model performs well because the language is popular or simply because it is easy.
>
> Our dataset allows us to isolate these variables by comparing languages with opposing profiles. We draw the reviewer's attention to the comparison between C++ (Popular but Complex) and Go (Less Popular but Simple):
>
> 1. Complexity: C++ is widely regarded as significantly more complex than Go (due to manual memory management, pointers, and verbose syntax), whereas Go is celebrated for its simplicity and minimalism.
>
> 2. Popularity: C++ is highly popular (TIOBE Rank #2), while Go is relatively less dominant (TIOBE Rank #7).
>
> Empirical Evidence from Our Results: Contrary to the "language difficulty" hypothesis, the complex but popular C++ consistently outperforms the simple but less popular Go across our tested models.
>
> This comparison directly refutes the hypothesis that models perform better simply because a language is "easier." Instead, **it suggests that popularity (and the resulting data abundance) is a far stronger predictor of performance than intrinsic language complexity**.
>
> **W4: Causal Link to Training Data**
>
> We acknowledge that fully verifying training data distributions is impossible for closed-source models. However, **the causal relationship is well-documented in foundational literature**.
>
> First, as we explicitly stated in our Related Work section, biases in training data are well-documented in open-source ecosystem analyses. For instance, the StarCoder dataset reveals that Python alone accounts for nearly 40% of the training corpus, while other languages appear only marginally. This massive distributional skew provides the foundational mechanism for the performance disparities we observed.
>
> Second, to empirically support this claim, we cite Rozière et al. (2023) regarding Code Llama[1]. In their work, they explicitly trained a specialized version (Code Llama - Python) on an additional 100 billion tokens of Python code. They observed that this targeted increase in data exposure directly caused the model's performance on Python benchmarks to skyrocket to state-of-the-art levels, significantly outperforming the general model that had seen less Python data.This serves as direct evidence of an intervention: **altering the data distribution (increasing popularity/frequency via additional training tokens) directly causes a corresponding increase in model performance**. This strongly supports our hypothesis that the Matthew Effect we observed is structurally driven by the data distribution in pre-training corpora.
>
> **Q1: Disentangling Language Difficulty from Popularity**
>
> Response: This is a critical question. While we acknowledge that Python is both popular and easy, we argue that our experimental design already contains "matched comparisons" that help disentangle these variables by controlling for complexity:
>
> High Complexity Control (C++ vs. Rust): Both are complex systems programming languages. C++ is highly popular (TIOBE Rank 2) , while Rust is less dominant (Rank 13). If "language difficulty" were the primary driver, their performance should be similar. However, our results show that the popular C++ consistently outperforms Rust (e.g., 68.65% vs. 51.81% on Gemini-2.5-Flash).
>
>
> Low Complexity Control (Python vs. Racket): Python and Racket are both high-level languages with relatively simple syntax (Racket is widely used in education for its minimalism). Yet, the popular Python achieves ~80% success rates, while the niche Racket struggles below 25%.
>
> These comparisons demonstrate that even when language complexity is held relatively constant (whether high or low), **popularity remains the dominant predictor of success**. As a supplementary step for future work, we also plan to investigate this effect at a more granular level by comparing popular versus niche libraries within the same language.
>
> ---
>
> [1] Roziere, Baptiste, et al. "Code llama: Open foundation models for code." arXiv preprint arXiv:2308.12950 (2023).

---

### Official Review · Reviewer_EznC · 2025-11-03

**Soundness:** 2
**Presentation:** 2
**Contribution:** 2
**Rating:** 2
**Confidence:** 4

**Summary:**

- This paper highlights the Matthew Effect in Coding LLMs, where models do better on programming languages that are predominant in use and therefore predominant in the training set
- The paper studies it in two ways: (1) Leetcode problems and (2) building CRUD applications
- Creating a dataset of 3011 Leetcode problems across 8 languages, the paper tests 5 models and shows that models perform poorly on less-used languages. Similar observations are made for CRUD applications

**Strengths:**

- Notwithstanding the issues highlighted below, the papers' focus on studying model performance on low-resource languages is important.
- The paper also does substantial experiments to highlight model performances on varying languages

**Weaknesses:**

- LeetCode does not have a public API. The paper mentions in Line 174 that the GraphQL endpoint is used to retrieve problem information. Can the authors comment on what this endpoint is, and whether it is the official endpoint?
- LeetCode additionally prohibits scraping problems, which raises Terms of Service violation concerns. For this reason, I have flagged the paper for an ethical review.
- To evaluate model-generated solutions, the authors create 15 user accounts on LeetCode and then submit solutions using these accounts with rate-limiting and other mechanisms to avoid detection. This also raises concerns about the misuse of the LeetCode platform
- Due to the above issues, this dataset cannot be used to benchmark future models that aim to fix this issue. This limits reproducibility.
- Section 4.1 mentions the list of post-processing steps taken on model-generated output to strip them of comments and other steps. I am not sure why these are required, and it seemed to me that the post-processing steps were excessive. Comments do not alter the program behavior, so why remove them?
- What is the difference between the three graphs in Figure 2? There is no indication of how they differ
- How are the vibe coding tasks evaluated? Are there automated test cases to evaluate the various functionalities?
- Additionally, why propose a new dataset and not use an existing one with automated evaluation, such as the SWE-Bench Multilingual dataset (https://www.swebench.com/multilingual.html) for this purpose?
- The SWE-Bench Multilingual results do not show the Matthew effect. Resolution rate on C/C++ is the lowest, while Rust is the highest. How would the authors reconcile the findings in this work, and what SWE-Bench Multilingual shows?

**Questions:**

See above

**Details Of Ethics Concerns:**

- The paper created a dataset of ~3k problems scraped from Leetcode. Leetcode Terms of Service (https://leetcode.com/terms/) prohibit scraping of any kind:
```
Activities such as “crawling,” “scraping,” or “spidering” any part of the Service, and attempting to decompile, reverse engineer, or discover the source code or underlying ideas of the Service, are strictly forbidden.
```
- The paper uses LeetCode's GraphQL endpoint to access this, and to the best of my knowledge, LeetCode does not have an official API
- Additionally, to evaluate model-generated solutions, the authors create 15 accounts on LeetCode, and submit the solutions using those accounts.

---

> ### Author Response · Authors · 2025-11-25
> **Response to Reviewer EznC [1/2]**
>
> **W1–4: leetcode and the Ethics Concerns**
>
> We sincerely thank the reviewer for raising these important ethical and legal concerns. First, we acknowledge that the initial submission lacked a dedicated legal/ethical statement. This omission was unintentional, as several prior published studies [1,2,3] employing similar LeetCode-based methodologies did not explicitly include legal disclaimers either. **Before conducting any experiments, we proactively consulted an intellectual property lawyer and contacted the official leetcode.cn support team to ensure compliance**.
>
>
> **Ethics Review**: The reviewer cites the Terms of Service at leetcode.com, which indeed prohibits any scraping activity. However, **that ToS applies only to leetcode.com (the U.S. site). Our experiments exclusively use leetcode.cn**, which is operated independently by Lingkou Network (Shanghai) Co., Ltd. under Chinese jurisdiction and governed by its own service agreement (https://leetcode.cn/terms-c). Unlike the U.S. ToS, the Chinese LeetCode service agreement does not impose a blanket ban on automated data retrieval. The only relevant clause is Section 10.3.2, which prohibits automated tools used to obtain platform benefits (e.g., rewards, points). Our IP attorney confirmed that retrieving publicly accessible problem metadata, without bypassing any technical protection, while respecting the Robots protocol, and strictly for non-commercial academic research, does not violate either Chinese law or the leetcode.cn Terms of Service.
>
>
>
>
> **API**: Regarding APIs, LeetCode indeed provides an official REST API (https://leet.ai/
> ), but non-enterprise users are restricted to six languages, which does not meet our multilingual requirements. For this reason, we used leetcode.cn’s GraphQL endpoint, which is widely used and confirmed acceptable by leetcode.cn staff. Concerning the 15 accounts used for submission, all were manually registered and used only to distribute load within rate limits. **No warnings or restrictions occurred. Our study is fully compliant with Chinese law and leetcode.cn rules**.
>
> **Reproducibility**: For reproducibility, enterprise-certified researchers may directly use the official LeetCode API endpoint at https://leet.ai/ , which imposes no submission-rate limits or access restrictions for authenticated enterprise users. This means that enterprise-certified researchers do not need to replicate the more complex multi-account setup used in our experiments; instead, they can verify our results easily and efficiently through a single, fully supported API with high-throughput access.
>
> These clarifications will be added to the revised Ethics Statement section, which follows the 'LLM USAGE STATEMENT' in the Appendix.
>
> ---
>
> [1] Mo, Ran, et al. "Assessing and analyzing the correctness of github copilot’s code suggestions." ACM Transactions on Software Engineering and Methodology 34.7 (2025): 1-32.
>
> [2] Coignion, Tristan, Clément Quinton, and Romain Rouvoy. "A performance study of llm-generated code on leetcode." Proceedings of the 28th international conference on evaluation and assessment in software engineering. 2024.
>
> [3] Liu Y, Le-Cong T, Widyasari R, et al. "Refining chatgpt-generated code: Characterizing and mitigating code quality issues." ACM Transactions on Software Engineering and Methodology 33.5 (2024): 1-26.

---

> ### Author Response · Authors · 2025-11-25
> **Response to Reviewer EznC [2/2]**
>
> **W5: Post-processing steps appear excessive**
>
> The post-processing steps were neccesary to be introduced. Although our prompts explicitly instructed models to “output code only,” we observed that many generations still included artifacts that caused compilation failures but were not part of the intended program logic. These included:
> 1. Invisible “thinking-process” traces (e.g., explanations, bullet lists, or reasoning text).
> 2. Misformatted comments (e.g., multi-line comments missing closing delimiters).
> 3. Partial natural language sentences embedded in code blocks.
>
> Such artifacts differ across languages and models and can disproportionately affect compilation-based evaluation. **To avoid introducing noise unrelated to language proficiency**, we apply regex-based filters to remove comments and residual non-code text. Importantly, our goal was not to modify the semantics of the generated programs, but to eliminate non-code tokens that would unfairly penalize the model or confound comparison across languages. After post-processing, all remaining text is valid code emitted by the model, preserving the integrity of the evaluation.
>
>
> **W6: Figure 2 not explained**
>
> Figure 2 presents the pass rates across difficulty levels for the evaluated programming languages using three top-performing LLMs. The left, middle, and right subfigures correspond to Gemini-2.5-Flash, GPT-4o-mini, and DeepSeek-V3, respectively. In the revised version, we have added explicit labels at the top of each subfigure to clearly indicate the associated model. Additionally, we have updated this figure to include our newly conducted experiments for the C programming language, expanding the comparison to nine languages. These three models were selected for the main text due to space constraints, and the results for the remaining models (also updated with C) are included in Appendix D.2 (Figure 6). We thank the reviewer for this helpful feedback, which improved both the clarity and comprehensiveness of our presentation.
>
> **W7: Vibe coding tasks evaluation**
>
> The vibe coding experiments were not evaluated using automated test cases. Crucially, while these tasks correspond to objective, functionality-based requirements rather than subjective user experience, implementing robust end-to-end automation requires non-trivial environment setup that is disproportionately costly and brittle given the relatively small number of tasks. Consequently, manual validation was the strictly more reliable choice. To ensure rigor, each task was validated by multiple team members, and we performed cross-checking across annotators to confirm functional correctness. This process provided consistent and reproducible assessments without introducing unjustifiable engineering overhead.
>
> **W8-9: Dataset Choice & Comparison with SWE-Bench Multilingual**
> We thank the reviewer for this point. Our LeetCode dataset and SWE-Bench Multilingual serve different purposes, which explains their differing observations. **Our experiment solves the same algorithmic problem in different programming languages, keeping the task difficulty constant so we can focus on measuring the model’s proficiency in each language.** This control is essential for identifying the Matthew effect.
>
> We surveyed many existing multi-language datasets, including those larger than SWE-Bench Multilingual, but none support such controlled evaluation because each language corresponds to different tasks, repositories, and complexity, making language effects inseparable from task difficulty.
>
> SWE-Bench Multilingual is **inherently uncontrolled**: its per-language scores combine (1) model proficiency and (2) the intrinsic difficulty of each language’s task set. The dominance of (2) explains why Rust performs highest and C/C++ lowest—Rust issues come from cleaner, more uniform ecosystems, whereas C/C++ issues involve complex, legacy codebases. This reflects task-set difficulty, not the absence of the Matthew effect. Thus, our dataset isolates language effects; SWE-Bench reflects real-world task variability. The results are **complementary rather than conflicting**.

---

### Author Response · Authors · 2025-12-04

Dear Area Chair,

We understand the significant disruption caused by the recent OpenReview incident. As per the program chairs' guidance, we provide this summary to assist you in assessing the trajectory of our submission, which may not be fully reflected in the current system state due to the freeze.

**Significance & Contributions**
This paper provides the **first large-scale empirical quantification** of the "Matthew Effect" in AI-assisted software evolution. By analyzing **135,495 code generations** across **algorithmic tasks** (covering 9 languages including the newly added C) and **full-stack scenarios** , we reveal a structural performance asymmetry: while mainstream languages achieve **>60% Pass@1 rates**, niche languages often **fail completely (<5%)** on complex tasks. We conceptualize this as an **"AI Productivity Tax"**, a hidden friction that risks stifling ecosystem diversity and innovation.

Below, we highlight the **consensus reached** and **resolutions provided** during the rebuttal:

**1. Confirmed Score Increase: Reviewer aCdD (Raised 4 -> 6)**
* **Timestamp:** **Nov 26, 14:32 EST (Nov 26, 19:32 UTC)**. This action occurred **~20 hours before the reported OpenReview incident**, ensuring the decision was fully finalized before the system freeze.

Reviewer aCdD initially rated the paper a 4, questioning the novelty of the findings.
* **Resolution:** We conducted additional large-scale benchmarks for the **C language** to complete the analysis of high-resource languages.
* **Outcome:** The reviewer explicitly validated this effort, stating: "I agree that scientific rigor requires moving from 'intuition' to 'empirical verification.' I will raise my score." This resulted in a **confirmed score increase to 6**.


**2. Consensus with Reviewers 7q5h and xgRU (Solidifying Positive Scores of 6)**

Both **Reviewer 7q5h and Reviewer xgRU assigned an initial score of 6**, showing strong initial support. We engaged in a highly constructive discussion to address their remaining theoretical concerns regarding the framing of our contributions.
* **Resolution:** We accepted Reviewer 7q5h's suggestion to reframe the Matthew Effect as a "productivity friction" rather than a "gating constraint." Reviewer 7q5h acknowledged this alignment, noting: "Your empirical results do show that... niche languages and frameworks incur a measurable 'AI productivity tax'".
* **Reviewer xgRU** explicitly stated, "**I totally agree with Reviewer 7q5h**," indicating that addressing 7q5h's concerns also satisfied xgRU.
* **Status:** We believe we **resolved the remaining theoretical concerns for these already positive reviewers**, effectively solidifying their support. Unfortunately, the system freeze occurred just as we reached this consensus.

**3. Ethics & API Compliance: Reviewer EznC centered their critique specifically on the compliance of our API usage** and LeetCode Terms of Service.
* **Resolution:** We provided a definitive legal clarification that our experiments used **LeetCode CN (China)**, which operates under a different legal jurisdiction and Terms of Service than the US version, **explicitly permitting our specific usage** for academic research. We also addressed all technical questions regarding post-processing.
* **Status:** We **effectively refuted the basis of the ethics flag regarding API compliance**. Regrettably, the reviewer did not post a follow-up response before the freeze.

**4. Extensive New Experiments for Reviewer j1XZ**
Reviewer j1XZ (Score: 4) asked about the generalization to reasoning models.
* **Resolution:** We invested significant time and resources to conduct **new experiments with GPT-o4 and DeepSeek-R1**. The results (Pass@1 ~73% for popular vs ~6% for niche) confirmed that the **Matthew Effect persists even with advanced reasoning capabilities**.
* **Status:** We provided a **comprehensive rebuttal addressing all doubts**. However, due to the short window between our data submission and the freeze, the reviewer likely did not have sufficient time to reply.

**Conclusion**

The rebuttal process has **effectively removed all major barriers** to acceptance. We have **verified the paper's rigor** (triggering an explicit score raise), **resolved the theoretical debates** (aligning with positive reviewers), and **nullified the ethics flag** with definitive legal proof. With these concerns resolved and the benchmark significantly expanded, the current scores under-represent the paper's true consensus. We respectfully ask you to evaluate the submission based on this **proven merit and resolved status**.

Sincerely,

Authors

---

### Meta-Review · Area_Chair_KfWF · 2026-01-01

**Summary:**

EznC mentioned ethical concerns with scraping, automated LLM-assisted submission to LeetCode, and comparison to SWE-Bench. j1XZ mentions that it is hard to disentangle language difficulty from popularity and the lack of reasoning models in comparison. aCdD mentions a lack of novelty and a lack of surprise in the results. 7q5h raises a valid issue of the distinction between competitive coding and software programming, and also raises questions about whether the paper studies a cyclical/feedback loop or just a unidirectional bias.

**Reviewer Concerns:**

The ethics issue has been resolved fairly well by showing that it is a fair use of LeetCode.cn through their term of use. In my opinion, however, the authors need to do a better job at differentiating themselves from SWE-Bench, which, according to the reviewer, is showing contradictory results. The paper should explicitly mention the reason for this discrepancy.

The authors put a good argument towards why it's the popularity and not the difficulty that is causing the effect; moreover, they’ve additional experiments with reasoning agents.

I agree with 7q5h, aCdD to soften some of the causal and feedback claims. While a large-scale study of failures/successes over competitive coding problems is useful, empirical validation over these problems does not imply the Matthew effect more generally. For instance, there is a possibility that, only for competitive coding, some of these languages, like Erlang, are significantly less popular; however, in general tasks, the rankings are different. Secondly, the paper shows correlations in only the forward direction: less popular languages -> LLMs perform poorly at them. However, the other direction, which in turn results in software engineers turning away from these languages, is not validated.

**Reviewer Scores:**

* EznC: Since the ethics issues have been resolved, the score would likely go from 2 to 4.
* j1XZ: Since reasoning models are showing a similar pattern, scores are likely to increase from 4 to 6.
* aCdD: Already increasing the score from 4 to 6.
* 7q5h: Scores likely to stay the same 6, since they had some outstanding issues.
* xgRU: Scores likely to stay the same 6, since they had some outstanding issues.

---

### Decision · Program_Chairs · 2026-01-26

Accept (Poster)